# Engineering microbial division of labor for plastic upcycling

Teng Bao[1,2,9], Yuanchao Qian[1,2,9], Yongping Xin[1,2,9], James J. Collins [3,4,5] ✉ & Ting Lu [1,2,6,7,8] ✉

Plastic pollution is rapidly increasing worldwide, causing adverse impacts on the environment, wildlife and human health. One tempting solution to this crisis is upcycling plastics into products with engineered microorganisms; however, this remains challenging due to complexity in conversion. Here we present a synthetic microbial consortium that efficiently degrades polyethylene terephthalate hydrolysate and subsequently produces desired chemicals through division of labor. The consortium involves two *Pseudomonas putida* strains, specializing in terephthalic acid and ethylene glycol utilization respectively, to achieve complete substrate assimilation. Compared with its monoculture counterpart, the consortium exhibits reduced catabolic cross-talk and faster deconstruction, particularly when substrate concentrations are high or crude hydrolysate is used. It also outperforms monoculture when polyhydroxyalkanoates serves as a target product and confers flexible tuning through population modulation for *cis-cis* muconate synthesis. This work demonstrates engineered consortia as a promising, effective platform that may facilitate polymer upcycling and environmental sustainability.

Plastics are foundational materials that support our modern society. However, their remarkable versatility and limited recyclability have caused a growing problem of plastic pollution, adversely impacting the environment, wildlife, and human health worldwide[1–3]. Recently, microbial-based bioconversion is emerging as an attractive alternative to mechanical, thermal, and chemical strategies for polymer upcycling, with the potential to simplify processing procedures and integrate waste degradation with product generation[4–7]. Using polyethylene terephthalate (PET)—one of the most common plastics—as an example, a variety of microbial isolates such as *Ideonella sakaiensis*, *Thermobifida fusca*, and *Saccharomonospora viridis* have been discovered and harnessed to break down and assimilate the polymer[8–10]. Concurrently, enzymes are increasingly optimized through random mutagenesis and machine learning-guided engineering to enhance deconstruction

efficiency[11–14]. Additionally, different bacterial and fungal strains have been engineered to accelerate PET hydrolysis as well as to convert plastic waste into valuable chemicals and products[15–20].

Despite these exciting developments, current biotransformation focusing on monocultures faces several challenges owing to the complexity of polymer upcycling. First, complete PET depolymerization yields a mixture of terephthalic acid (TPA) and ethylene glycol (EG) that are heterogeneous in their physiochemical characteristics. In the presence of such mixed substrates, carbon catabolite repression between metabolic pathways is a ubiquitous nature of microorganisms that makes it difficult to simultaneously utilize multiple substrates[21,22]. Second, depolymerization products such as PET hydrolysate are often inhibitory to microbes and can impair cellular metabolism for efficient and complete substrate assimilation, particularly when the products

[1]Department of Bioengineering, University of Illinois Urbana-Champaign, Urbana, IL, USA. [2]Carl R. Woese Institute for Genomic Biology, University of Illinois Urbana-Champaign, Urbana, IL, USA. [3]Department of Biological Engineering and Institute for Medical Engineering & Science, Massachusetts Institute of Technology, Cambridge, MA, USA. [4]Wyss Institute for Biologically Inspired Engineering, Harvard University, Longwood, MA, USA. [5]Broad Institute of MIT and Harvard, Cambridge, MA, USA. [6]Center for Biophysics and Quantitative Biology, University of Illinois Urbana-Champaign, Urbana, IL, USA. [7]Department of Physics, University of Illinois Urbana-Champaign, Urbana, IL, USA. [8]National Center for Supercomputing Applications, Urbana, IL 61801, USA. [9]These authors contributed equally: Teng Bao, Yuanchao Qian, Yongping Xin. ✉e-mail: jimjc@mit.edu; luting@illinois.edu

are abundant or crude hydrolysate is used. Third, to upgrade plastic waste into valued products, strains need to encode advanced designer pathways composed of multiple enzymes and steps to yield intermediates and final products, thereby demanding system-level fine-tuning that is complicated for single strains.

We reasoned that designer microbial consortia could serve as a possible solution to address these challenges. Toward functionality programming, ecosystem-based microbial engineering has been recently proposed as a versatile strategy that is potentially superior to single-strain construction[23–27]. Lying at the heart of microbial consortia is the division of labor (DOL), a form of organization through which a goal is divided into subtasks among constituting members[23,28]. For plastic upcycling, in principle, DOL promises to reduce catabolic interactions ubiquitous in the presence of mixed substrates by partitioning assimilation pathways among individual members. It also allows each strain to specialize in one designated task without compromising its specialty for fulfilling multiple purposes and to bear a lower level of stress, thereby helping cells to maintain metabolic homeostasis for better functioning. Moreover, by compartmentalizing subsystems, DOL promotes the orthogonality of individual modules, which facilitates systematic fine-tuning and optimization of engineered pathways. Indeed, recent studies have exemplified these arguments and harnessed synthetic consortia for applications under different settings[29–36].

In this study, we tested the utility of DOL for PET upcycling by means of ecosystem engineering and analysis. We designed a synthetic microbial consortium consisting of two *Pseudomonas putida* strains, which individually specialize in TPA and EG assimilation, respectively, but collectively achieve complete consumption of PET hydrolysate. For comparison, we also built a single strain for TPA and EG co-consumption. We compared the efficiency of this pair of engineered systems under various substrate conditions and extended them to upcycle PET by adopting medium chain length polyhydroxyalkanoates (mcl-PHA) and *cis-cis* muconate (MA) as value-added products.

Together, our study showcases the potential of engineered consortia for plastic degradation and conversion, shedding light on the biological mitigation of growing plastic waste.

## Results

### Construction of a synthetic microbial consortium for TPA and EG co-consumption

We started by building a *P. putida* consortium involving two strains that specialize in TPA and EG assimilation, respectively (Fig. 1a). Here, the soil bacterium *P. putida* was selected as the cellular chassis due to its versatile metabolic capacity, genetic amenability for manipulation, and prior demonstration on valued product production from waste-derived feedstocks. The TPA specialist (Pp-T) was developed from the EM42 strain by abolishing its EG assimilation and enabling TPA consumption. The former was achieved by deleting the entire *ped* gene cluster from the EM42 genome to eliminate its EG oxidation for glycolate generation[37,38]. The latter was implemented by introducing into the chassis a *tpa* cluster from *Rhodococcus jostii* RHA1[39], which encodes the large and small subunits of terephthalate 1,2-dioxygenase (*tpaAa* and *tpaAb*), a reductase component (*tpaB*), a dihydrodiol dehydrogenase (*tpaC*) and a transporter (*tpaK*), to confer the conversion of TPA to protocatechuate (PCA) for strain utilization. The EG specialist (Pp-E) was constructed from M31, an EG-assimilating strain derived from EM42, by deleting the transcriptional repressor gene (*gclR*) of the Gcl pathway[18,38] to enhance the carbon flux from glyoxylate to pyruvate, and replacing the native promoter and ribosomal binding site (RBS) of the glycolate oxidase (*glcDEF*) operon with a strong constitutive promoter ($P_{tac}$) and an artificial RBS[40] to overcome glycolate accumulation[41]. We hypothesized that such a DOL design would yield catabolically orthogonal specialists that individually consume TPA or EG but collectively enable simultaneous TPA and EG utilization when co-cultured as an ecosystem (T-E consortium).

To test this hypothesis and validate the consortium, we performed batch fermentations using TPA, EG, and their mix as substrates.

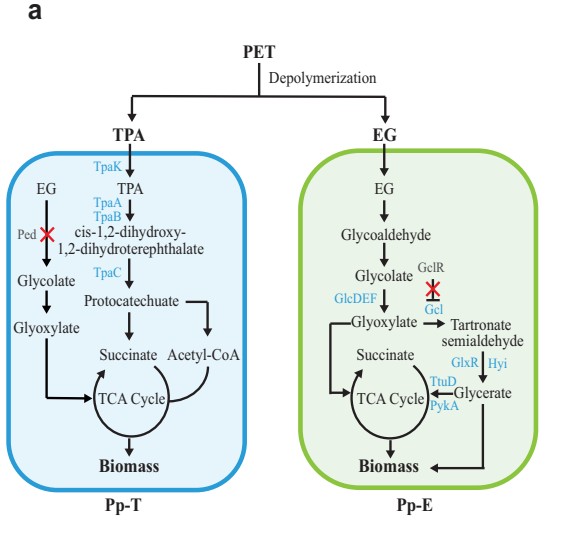

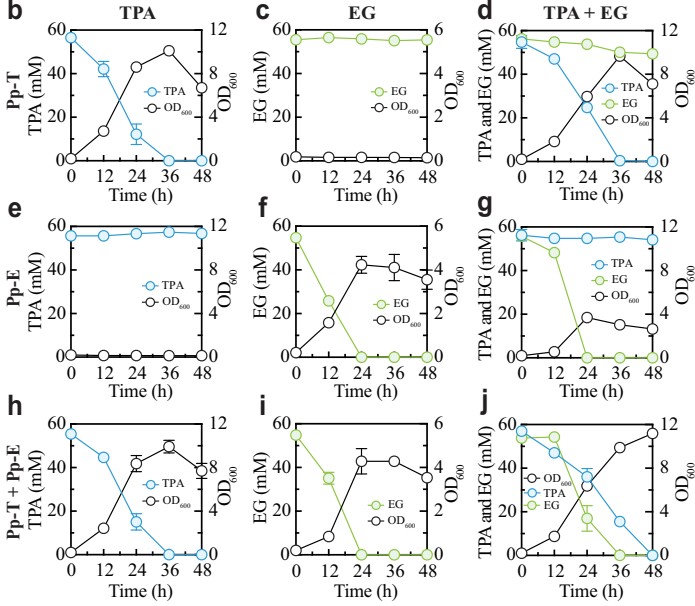

**Fig. 1 | A synthetic *P. putida* consortium for simultaneous TPA and EG degradation. a** Schematic of the designer T-E consortium composed of two strains Pp-T and Pp-E. Pp-T specializes in TPA degradation, which was developed by deleting the *ped* operon and constitutively expressing the genes *tpaAa, tpaAb, tpaB, tpaC*, and *tpaK*. By contrast, Pp-E specializes in EG degradation, achieved by knocking out the *gclR* gene to boost the expression of the genes *gcl, glxR, hyi, ttuD*, and *pykA* and replacing the native promoter of the *glcDEF* operon with a strong constitutive promoter ($P_{tac}$). **b–d** Temporal profiles of substrate degradation and cell growth during TPA (**b**), EG (**c**), and TPA and EG mixture (**d**) fermentations by Pp-T. **e–g** Temporal profiles of substrate degradation and cell growth during TPA (**e**), EG (**f**), and TPA and EG (**g**) fermentations by Pp-E. **h–j** Temporal profiles of the TPA (**h**), EG (**i**), and TPA and EG (**j**) fermentations by the T-E consortium. The co-culture was inoculated at a 1:1 ratio. For panel **a**, blue text indicates overexpressed genes whereas red cross ('×') indicates the deletion of genes in dark gray. Experimental data are presented as mean values with standard deviations from three independent experiments. Source data are provided as a Source Data file.

Indeed, Pp-T successfully utilized TPA ($Na_2$TPA) as the sole carbon source for growth and depleted 56.2 mM TPA within 36 h (Fig. 1b). The strain also completely consumed at least 100 mM TPA and grew in the presence of 316 mM TPA, suggesting a high level of TPA tolerance (Supplementary Fig. 1a). Meanwhile, Pp-T did not degrade EG when it served as a sole substrate (Fig. 1c). When both TPA and EG were present, the former was fully utilized in 36 h while the latter remained largely unchanged (Fig. 1d and Supplementary Fig. 1b), confirming that Pp-T specializes in TPA assimilation. Notably, unpinning this specialization is our deletion of the *ped* cluster responsible for EG oxidation. To confirm the functional role of the specialization, we compared the growth, TPA consumption, and intermediate metabolites of Pp-T and its parental strain, Pp-$T_0$, which possesses the native *ped* cluster and heterologous *tpa* cluster (Supplementary Fig. 2). We found Pp-T remained efficient in TPA assimilation and biomass accumulation with the increase of EG concentrations for a given TPA level (100 mM) as compared to Pp-$T_0$ which was much more sensitive to the presence of EG. Accordingly, Pp-T yielded profiles of related metabolites different from Pp-$T_0$. These results suggested that enhancing the specialization of TPA degradation in Pp-T greatly reduces the catabolite interference caused by EG. Thus, Pp-T showed encouraging performance metrics with respect to TPA utilization and catabolic orthogonality.

On the other hand, Pp-E did not consume TPA (Fig. 1e), but was effective in utilizing EG as a sole carbon source (Fig. 1f). Compared to its parental strains, Pp-E was greatly improved in growth and EG consumption along with a negligible intermediate accumulation (Supplementary Fig. 3a). Pp-E also tolerated a high level of EG (Supplementary Fig. 3b), and exhibited a high specificity in EG consumption when TPA and EG were both supplied (Fig. 1g). Of note, Pp-E's specialization in EG assimilation and orthogonality to TPA consumption is attributed to the fact that Pp-E does not have a TPA catabolic pathway. Meanwhile, for a given EG level (e.g., 56.2 or 100 mM), the EG assimilation and biomass growth of Pp-E both decreased and were eventually abolished with the increase of TPA (Supplementary Fig. 3c, d), indicating that Pp-E is sensitive to TPA, different from Pp-T which is tolerant to EG (Supplementary Fig. 1b).

We also examined the co-culture of Pp-T and Pp-E for substrate utilization. In the presence of TPA or EG alone, the consortium showed similar fermentation patterns as those of the Pp-T and Pp-E monocultures (Fig. 1h, i). However, when TPA and EG were both supplied, the consortium efficiently and fully catabolized both substrates along with significant biomass production in 48 h (Fig. 1j), distinct from either of the monocultures (Fig. 1d, g). The results demonstrate our successful development of the TPA and EG co-utilization consortium.

## Comparison of the T-E consortium and its single-strain counterpart for substrate co-utilization

To determine whether co-culture could be superior to monoculture for the consumption of PET monomers, we engineered a *P. putida* strain (Pp-TE) that utilizes both TPA and EG (Fig. 2a). Here, the dual catabolic ability was programmed by loading the *tpa* cluster, used for the TPA specialist Pp-T, into the EG specialist Pp-E. Subsequent experiments confirmed that the resulting strain, Pp-TE, was able to grow on TPA or EG as a sole substrate (Fig. 2b, c). However, compared with Pp-T (Fig. 1b), Pp-TE was slower in TPA-to-biomass conversion and took 48 h to deplete 56.2 mM TPA (Fig. 2b). To examine if this is a generic trait of Pp-TE, we carried out single-substrate fermentations with various substrate concentrations. We found that the lag time increased systematically with the TPA level and, at 316 mM, the TPA consumption fully stopped (Supplementary Fig. 4a), suggesting that Pp-TE has a lower TPA tolerance than Pp-T (Supplementary Fig. 1a). By contrast, a high level of EG did not impair Pp-TE for EG assimilation (Supplementary Fig. 4b), similar to the case of Pp-E (Supplementary Fig. 3b); however, a longer lag time and hence a slightly slower consumption were observed during Pp-TE's EG assimilation, particularly when the EG concentration was high (Supplementary Figs. 3b, 4b).

We next assessed the single utilizer, Pp-TE, and the T-E consortium with mixed TPA and EG fermentations. Different from the former which yielded a specified fermentation pattern under a given initial $OD_{600}$, the latter was variable in fermentation upon the alteration of its inoculation ratio (Supplementary Fig. 5, Fig. 2d, e). Specifically, varying the Pp-T:Pp-E ratio modulated not only substrate consumption, metabolite production, and biomass growth, but also the temporal dynamics of the community composition (Supplementary Fig. 5). While both Pp-TE and the T-E consortium completely degraded the two substrates (75 mM each), it took the consortium with its optimal inoculation (3:1 ratio) a much shorter time (62 h) than Pp-TE (75 h) (Fig. 2d, e). Thus, designing the specialist pair allowed us to

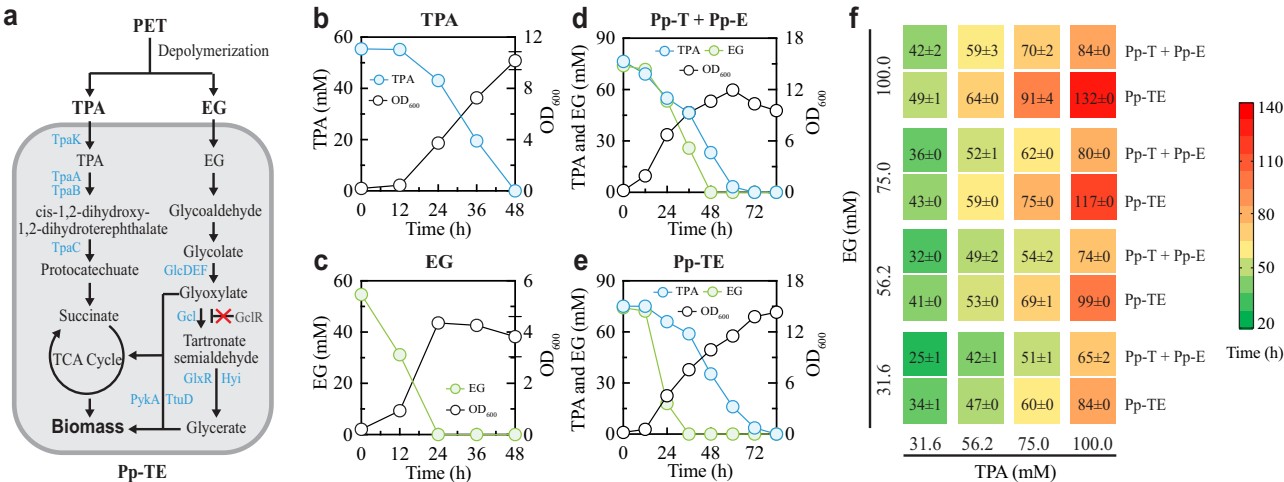

**Fig. 2 | Comparison of TPA and EG co-degradation by the T-E consortium and a single co-utilization strain. a** Schematic of a single strain (Pp-TE) capable of TPA and EG co-consumption. The entire *tpa* cluster was introduced and constitutively expressed in Pp-E to achieve TPA and EG co-degradation. **b** and **c** Temporal profiles of substrate degradation and cell growth during TPA (**b**) and EG (**c**) fermentations by Pp-TE. **d** and **e** Fermentation profiles of the T-E consortium (**d**) and Pp-TE (**e**) during a mixed TPA and EG fermentation. The consortium was inoculated at a 3:1 ratio for Pp-T and Pp-E. **f** Comparison of the degradation times of the T-E consortium and the single strain Pp-TE for various TPA and EG mixtures. The color of each grid indicates the time needed to fully degrade the corresponding mixture, which is also directly shown on the grid. For panel **a**, blue text indicates overexpressed genes whereas red cross ('×') indicates the deletion of genes in dark gray. Experimental data are presented as mean values with standard deviations from three independent experiments. Source data are provided as a Source Data file.

enhance the overall consumption efficiency of the mixed substrates. Additionally, we noticed a difference in assimilation kinetics: TPA and EG were consumed simultaneously by the T-E consortium; by contrast, in the case of Pp-TE, EG was rapidly utilized whereas TPA assimilation exhibited a significant delay that was alleviated only once the EG was mostly exhausted. This difference suggested that strong crosstalk exists between TPA and EG consumptions in Pp-TE, which was phenomenologically similar to sequential consumption in mixed sugar fermentation caused by carbon catabolite repression[22]. Meanwhile, for the same mixture, we observed a lower biomass accumulation by the T-E consortium than Pp-TE (Fig. 2d, e), which was likely owing to the higher production of intermediate metabolites, including a higher transient production of glycolate and a higher final accumulation of succinate, by the consortium (Supplementary Fig. 5c, k). This observation suggested that redirecting the succinate flux to desired biomass or product synthesis could be a potential consortium target for future optimization.

To ascertain the universality of the outperformance of the consortium in monomer assimilation, we expanded the scope of comparison by systematically examining the fermentation times and patterns of the T-E consortium and Pp-TE over 16 substrate combinations. As shown in Fig. 2f and Supplementary Fig. 6, although the two systems both successfully assimilated TPA and EG, the T-E consortium showed a systematic advantage in processing the mixture, especially when TPA alone or both substrates were abundant. For example, Pp-TE took 132 h to exhaust a mixture of 100 mM TPA and 100 mM EG but, for the consortium, it took only 84 h. Additionally, in this setting, the sequential consumption by Pp-TE became more noticeable in contrast to simultaneous co-utilization by the T-E consortium (the upper right

panels in Supplementary Fig. 6). The findings suggested that, by compartmentalizing the TPA and EG catabolic pathways, the T-E consortium has reduced catabolic cross-interactions between the pathways for a faster rate of assimilation compared to the generalist Pp-TE. Moreover, in theory, by assigning the pathways to separate strains, Pp-T and Pp-E each specializes in consuming one substrate without the need to compromise their specialty as Pp-TE. Thus, we demonstrated that the T-E consortium is an effective platform for TPA and EG co-consumption empowered by the engineered DOL.

## Complete PET degradation through chemical hydrolysis and microbial consumption

To apply our synthetic consortium for PET degradation, we proposed an integrated workflow that combines chemical hydrolysis with microbial deconstruction (Fig. 3a). Specifically, the workflow involves three steps. First, PET is depolymerized via alkaline hydrolysis. Second, the hydrolyzed product is mixed with M9 medium and filtered to form a growth medium. Third, the medium is inoculated with engineered microbes to fulfill fermentations. Although different strategies are available, we chose ester bonds hydrolysis with alkaline (NaOH) as the catalyst, because it does not require complex processing conditions[42,43] and an autoclave reactor is sufficient for performing the hydrolysis. In addition, the resulting products, disodium terephthalate (Na$_2$TPA, TPA salt) and EG can be directly used for biodegradation without extra procedures for acidification, separation, and purification[19].

To identify an optimal hydrolysis setting, we varied the NaOH concentration from 0% to 20% (Fig. 3b). Without NaOH (0%), the treatment did not result in any PET depolymerization. The TPA and EG

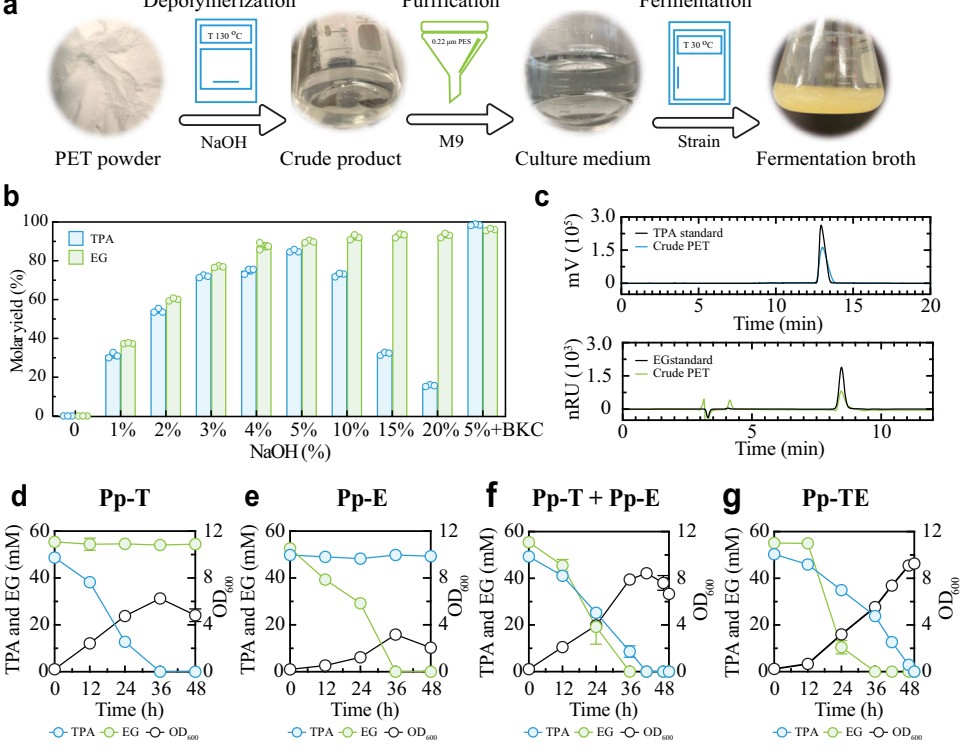

**Fig. 3 | Integrated chemical and microbial processing for PET deconstruction.** **a** Schematic illustration of an integrated PET processing workflow. PET is depolymerized through alkaline hydrolysis to generate a solution containing TPA and EG. The solution is then added with M9 medium to form a growth medium. Subsequently, the medium is inoculated with engineered microbes to fulfill fermentations to realize a full degradation. **b** The molar yield (%) of Na$_2$TPA and EG from PET powder under various hydrolysis conditions. **c** Characterization of the crude

hydrolysis product using pure TPA and EG as standard samples, confirming successful TPA and EG release from PET. **d–g** Temporal profiles of a direct PET hydrolysate fermentation by the Pp-T (**d**), Pp-E (**e**), T-E consortium (**f**), and the single strain Pp-TE (**g**). The Pp-T and Pp-E co-culture was inoculated at a 3:1 ratio. Experimental data are presented as mean values with standard deviations from three independent experiments. Source data are provided as a Source Data file.

molar yields from PET powder both increased with NaOH when its concentration was below 5%. However, when the concentration exceeded 5%, the Na$_2$TPA yield declined with the alkaline although the EG yield continued to rise minorly, likely due to the salting out of Na$_2$TPA in the presence of excess NaOH and a high level of EG[44]. Adopting 5% as the NaOH concentration, we further optimized the process by adding benzalkonium chloride (BKC) as a phase transfer catalyst, resulting in >97% of final yields for both TPA and EG. Using HPLC, we directly confirmed that the product was indeed composed of TPA and EG (Fig. 3c).

We proceeded to perform fermentations using the crude product (~50 mM TPA/EG) from the above hydrolysis. Figure 3d shows that Pp-T consumed TPA but not EG in the hydrolysate, at a rate comparable to the case of the pure TPA and EG mixture (Fig. 1d). Similarly, Pp-E depleted EG in the crude product (Fig. 3e); however, its rate was slower than the pure TPA-EG fermentation (Fig. 1g), suggesting the sensitivity of Pp-E to the hydrolysate. Additionally, we compared the fermentations by the T-E consortium and Pp-TE for different levels of the PET hydrolysate. At both the 31.6 mM (Supplementary Fig. 7a, b) and 56.2 mM (Fig. 3f, g) hydrolysate concentrations, the T-E consortium outperformed Pp-TE in terms of fermentation completion time, consistent with the observations for the pure mixture case (Fig. 2f and Supplementary Fig. 6). Meanwhile, at the 100 mM condition (Supplementary Fig. 7c, d), neither the consortium nor the single strain was able to survive and function, which is distinct from the pure mixture case whereby deconstruction was completed by the T-E consortium and Pp-TE in 84 h and 132 h, respectively (Fig. 2f). Again, this difference suggested that PET hydrolysate had increased toxicity compared to the pure TPA and EG mixture. Collectively, we established an integrated deconstruction process from powdered plastic to microbial biomass whereby the engineered consortium exhibited a compelling capability for substrate conversion.

## Upcycling of PET hydrolysate into mcl-PHA

To achieve upcycling, we further engineered our microbial consortia by designing and integrating metabolic pathways that upgrade the hydrolysate into valuable products. Specifically, we selected mcl-PHA, ranging from C6 to C14, as our target product because mcl-PHA is one of the most promising biodegradable polymers and the chassis organism *P. putida* has been shown to produce mcl-PHA from different substrates including sugars, organic acids, aromatic compounds and PET monomers[45,46]. For instance, engineered *P. putida* KT2440 was reported to produce 372 mg/L of mcl-PHA from pure EG under batch fermentation[41] and its relative, *Pseudomonas umsongensis* GO16, yielded 2349 mg/L of mcl-PHA from pure TPA in fed-batch fermentation[47] (Supplementary Table 3). However, the titer of *P. umsongensis* GO16 was dropped down to 150–210 mg/L when PET hydrolysate was used as cellular substrate[17,48], highlighting the need to investigate the co-assimilation of TPA and EG in the hydrolysate and explore alternative engineering strategies.

Building on the T-E consortium, we applied two parallel strategies to optimize mcl-PHA production (Fig. 4a). First, we overexpressed the genes *phaG* and *alkK*, which encode hydroxyacyl-ACP acyltransferase and acyl-CoA-synthase, respectively, to catalyze the conversion of (R)-3-hydroxyacyl-ACP to (R)-3-hydroxyacyl-CoA. The two enzymes are also essential for PHA biosynthesis under high nitrogen conditions and cell growth on aromatic compounds[49]. Therefore, we used a constitutive promoter P$_{EM7}$ to tandemly drive *phaG* and *alkK*, linked with two mcl-PHA synthases (*phaC1* and *phaC2*), in both Pp-T and Pp-E, resulting in the derivatives Pp-TP and Pp-EP, respectively. Second, we deleted from Pp-TP and Pp-EP *fadBA* and *fadBAxE* to limit the carbon flux from (R)-3-hydroxyacyl-CoA to acetyl-CoA, as well as the PHA depolymerase (*phaZ*) to eliminate PHA depolymerization[50], yielding new derivatives PpΔ-TP and PpΔ-EP, respectively. Similarly, we engineered the co-consumption strain Pp-TE (Fig. 4b), resulting in Pp-TEP

and PpΔ-TEP accordingly. For comparison, we additionally constructed three control strains—Pp-TS, Pp-ES and Pp-TES—by transforming the vector pSEVA421 into the corresponding parental strains, Pp-T, Pp-E and Pp-TE.

To evaluate these strains, we performed monoculture fermentations using an M9 medium supplemented with pure TPA and/or EG. During the TPA fermentation, Pp-TP and PpΔ-TP both produced more mcl-PHA (163.80 ± 16.83 and 140.96 ± 8.05 mg/L, respectively) with a higher yield (Pp-TP: 64.55 ± 4.32 mg$_{PHA}$/g$_{CDW}$ and 14.01 ± 1.37 mg$_{PHA}$/g$_{TPA}$; PpΔ-TP: 63.35 ± 0.44 mg$_{PHA}$/g$_{CDW}$ and 12.20 ± 0.57 mg$_{PHA}$/g$_{TPA}$) than their control Pp-TS (118.25 ± 5.37 mg/L, 48.12 ± 2.90 mg$_{PHA}$/g$_{CDW}$, and 9.72 ± 0.67 mg$_{PHA}$/g$_{TPA}$) (Fig. 4c and Supplementary Fig. 8j). However, they were 1.25 and 2.38 times slower in TPA depletion than Pp-TS (Supplementary Fig. 8a–c), likely owing to the genetic manipulations we implemented. Similarly, Pp-EP and PpΔ-EP exhibited a significant increase in mcl-PHA production (31.28 ± 1.57 and 34.82 ± 2.73 mg/L) and yield (Pp-EP: 20.79 ± 1.63 mg$_{PHA}$/g$_{CDW}$ and 8.69 ± 0.65 mg$_{PHA}$/g$_{EG}$; PpΔ-EP: 22.28 ± 1.64 mg$_{PHA}$/g$_{CDW}$ and 11.85 ± 2.80 mg$_{PHA}$/g$_{EG}$) compared with Pp-ES (3.39 ± 0.23 mg/L, 2.56 ± 0.17 mg$_{PHA}$/g$_{CDW}$ and 0.96 ± 0.06 mg$_{PHA}$/g$_{EG}$) during pure EG fermentation (Fig. 4d) but, accordingly, had an associated delay in EG depletion (Supplementary Fig. 8d–f). Notably, the absolute mcl-PHA yields of Pp-EP and PpΔ-EP were much lower than those of Pp-TP and PpΔ-TP (Supplementary Fig. 8j), owing to the lower carbon content in EG than in TPA and hence a lower carbon to nitrogen ratio (C:N ratio) in the growth medium, a critical regulator for mcl-PHA accumulation[51]. For single-strain co-utilization of pure TPA and EG mixture, Pp-TEP exhibited a higher titer (392.64 ± 15.70 mg/L) and a higher yield (127.76 ± 7.13 mg$_{PHA}$/g$_{CDW}$ and 28.01 ± 0.82 mg$_{PHA}$/g$_{TPA+EG}$) than the control Pp-TES (292.89 ± 12.37 mg/L, 98.00 ± 9.53 mg$_{PHA}$/g$_{CDW}$ and 20.95 ± 1.40 mg$_{PHA}$/g$_{TPA+EG}$), but PpΔ-TEP was lower in mcl-PHA titer (271.58 ± 23.75 mg/L) and yield (102.05 ± 3.05 mg$_{PHA}$/g$_{CDW}$ and 20.25 ± 2.67 mg$_{PHA}$/g$_{TPA+EG}$) (Fig. 4e and Supplementary Fig. 8j), even though the substrate degradation rates of Pp-TEP and PpΔ-TEP all reduced (Supplementary Fig. 8g–i). These experiments confirmed the phenotypes of our engineered strains and also revealed Pp-TP, Pp-EP, and Pp-TEP as the best upcycling candidates.

Next, we used the co-culture of Pp-TP and Pp-EP, named the TP-EP consortium, and the single-strain counterpart Pp-TEP to ferment PET hydrolysate for mcl-PHA production under nitrogen-limiting conditions. Figure 4f shows that the consortium simultaneously consumed TPA and EG at comparable rates, leading to the depletion of the hydrolysate in about 65 h. By contrast, Pp-TEP utilized all TPA but was significantly slower in EG degradation, resulting in significant EG remanent (82%) (Fig. 4g). For comparison, we repeated the fermentations under nitrogen-surplus conditions (Supplementary Fig. 9a, b), confirming that the consortium outperformed the single strain for hydrolysate assimilation regardless of nitrogen availability. In addition, comparison of Pp-TEP's fermentation characteristics with the pure TPA and EG mixture with nitrogen surplus (Supplementary Fig. 8h), hydrolysate with nitrogen surplus (Supplementary Fig. 9b), and hydrolysate with nitrogen limitation (Fig. 4g) confirmed that the EG assimilation capacity was greatly impaired by hydrolysate, consistent with recent reports on other monoculture-based fermentations of hydrolyzed PET[17,18,52], and further reduced upon nitrogen limitation. Associated with its advantage in substrate utilization, the TP-EP consortium showed a higher mcl-PHA titer and a higher yield than Pp-TEP in the presence or absence of nitrogen limitation (Fig. 4j, k, Supplementary Fig. 9c, d).

Additionally, to evaluate the stability and performance of the engineered systems, we performed fed-batch fermentation with pulse feeding. The TP-EP consortium continued to successfully deplete both TPA and EG at the hydrolysate feeding stage (Fig. 4h). By contrast, limited EG uptake of Pp-TEP led to an EG accumulation upon feeding, which reached 88.45 mM at the end of fermentation (Fig. 4i).

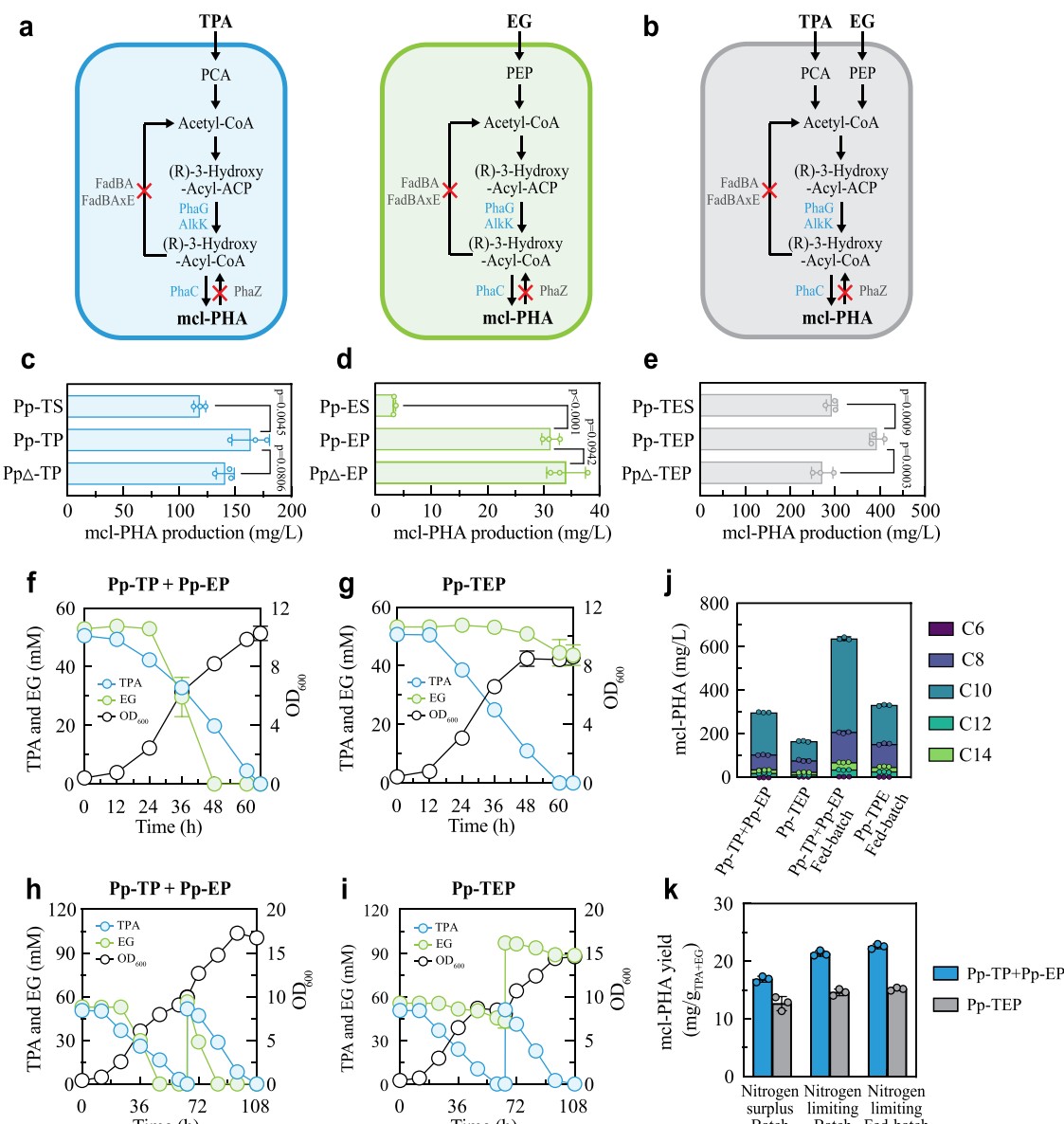

**Fig. 4 | Upcycling of hydrolyzed PET into mcl-PHA. a** Design of a synthetic consortium that produces mcl-PHA from TPA and EG co-degradation. The consortium was created by introducing an mcl-PHA biosynthetic pathway into Pp-T and Pp-E along with additional knockouts. **b** Design of an engineered strain that synthesizes mcl-PHA and co-utilizes TPA and EG. The strain was developed by adding an mcl-PHA synthesis pathway into Pp-TE along with additional knockouts. **c–e** Comparison of mcl-PHA production from pure TPA and EG by different Pp-T (**c**), Pp-E (**d**), and Pp-TE (**e**) derivatives. Pp-T has three derivates: Pp-TS, Pp-TP, and PpΔ-TP. Pp-E has three derivates: Pp-ES, Pp-EP, and PpΔ-EP. Pp-TE has three derivates: Pp-TES, Pp-TEP, and PpΔ-TEP. **f, g** Temporal profiles of direct PET hydrolysate fermentations under nitrogen-limiting conditions by the TP-EP consortium (**f**) and its single-strain counterpart Pp-TEP (**g**). An initial 1:3 inoculation ratio was used for Pp-TP and Pp-EP. **h, i** Temporal profiles of fed-batch fermentations under nitrogen-limiting conditions by the TP-EP consortium (**h**) and Pp-TEP (**i**). An initial 1:3 inoculation ratio was used for Pp-TP and Pp-EP. **j** The mcl-PHA titer and monomer composition of mcl-PHA produced by the TP-EP consortium and single-strain counterparts Pp-TEP in nitrogen-limiting batch and fed-batch cultures. **k**, The mcl-PHA yield from PET hydrolysate by the TP-EP consortium and Pp-TEP in different batch and fed-batch cultures. For panels **a** and **b**, blue text indicates overexpressed genes whereas red cross ('×') indicates the deletion of genes in dark gray. In panels **c–e**, **j**, and **k**, mcl-productions were calculated at the time point when TPA and/or EG were fully degraded. In panels **c–e**, results were compared using one-way ANOVA (one-sided) with Dunnett's multiple comparison test. Experimental data are presented as mean values with standard deviations from three independent experiments. Source data are provided as a Source Data file.

Subsequent analysis showed that the fed-batch of the TP-EP consortium and Pp-TEP both produced significantly more mcl-PHA than the single batch fermentation (Fig. 4j). Remarkably, through the fed-batch fermentation, the consortium produced 637.30 ± 10.14 mg/L mcl-PHA, which was about 92% higher than that of Pp-TEP. Compared to recent efforts (Supplementary Table 3), the TP-EP consortium represented a new advance in mcl-PHA titer by engineered organisms that directly and completely assimilated PET hydrolysate. A detailed monomer composition analysis of the product revealed that, although

the longer chain monomers such as 3-hydroxydodecanoate (C12) and 3-hydroxytetradecanoate (C14) exhibited a large increase, the majority of the product was 3-hydroxydecanoate (C10) and 3-hydroxyoctanoate (C8) which accounted for >75% of the total. The mcl-PHA yields in terms of substrate weight were largely comparable between the single and fed batch under nitrogen limitation, but higher than those in nitrogen surplus fermentation (Fig. 4k). Meanwhile, the yields in terms of cell dried weight were higher in fed batch and single batch (Supplementary Fig. 9d), likely due to reduced biomass

accumulation per substrate weight in fed batch and single batch. Thus, the consortium design presented two advantages over its single-strain counterpart for bioconversion – a faster and complete deconstruction of the hydrolysate and a higher production titer for mcl-PHA – showcasing the potential of engineered consortia for polymer upcycling.

## High-yield MA biosynthesis via DOL-mediated system optimization

In addition to reducing catabolic crosstalk, facilitating substrate co-consumption, and yielding a high level of production for the case of mcl-PHA as demonstrated above, microbial DOL promises effective optimization strategies for biological PET upcycling. Specifically, we argued that, by partitioning designer biosynthetic pathways among the members of a consortium, DOL would confer a system-level optimization of bioconversion through population modulation. To illustrate this point, we set out to upcycle the PET hydrolysate into *cis–cis* MA, an important chemical used to synthesize polyurethane and the nylon precursor adipic acid[53]. Recently, different efforts have been made for MA production (Supplementary Table 4). For example, a study reported a three-step process, involving PET hydrolysis, TPA and EG separation, and microbial catalysis, to convert TPA to MA; however, the resulting MA production was relatively low (0.38 g/L)[54]. Studies also showed a higher final titer of MA synthesis (4.59–73.8 g/L) by engineered *P. putida* but, in these cases, glucose was required as a sole substrate or as a supplementary carbon source[52,55–58]. Meanwhile, it has been reported that MA biosynthesis suffers from the accumulation of

intermediate metabolites[55,56]. One such intermediate is catechol (CAT), which can be converted from TPA and bioprocessed into MA. As CAT is toxic to microbes, its accumulation may lead to forced self-toxification[59], which impairs MA production. Thus, we designed a new ecosystem, termed TC-EM consortium, by introducing an additional layer of DOL into the T-E consortium to enable MA biosynthesis by using the PET hydrolysate as the sole carbon source without additional carbon supply.

As illustrated in Fig. 5a, the MA synthesis pathway was split into an upstream subsystem that converts TPA to CAT and a downstream subsystem upgrading CAT into MA. These two subsystems were then loaded into Pp-T and Pp-E, respectively, to generate Pp-TC and Pp-EM which constitute the TC-EM consortium. Specifically, Pp-TC was created by blocking the main CAT catabolic pathway by deleting the *catRBCA* cluster and introducing a codon-optimized PCA decarboxylase (*aroY*) and decarboxylase enhancer (*ecdB*) from *Enterobacter cloacae* ATCC 13047[60]. Pp-EM was constructed by deleting the *catRBC* operon and introducing the tac promoter ($P_{tac}$) to substitute the native promoter of the major catechol 1,2-dioxygenase (*catA*)[61] to boost the CAT-to-MA conversion.

Fermentations showed that Pp-TC successfully degraded 53.41 mM TPA over a course of 48 h during which CAT accumulated up to 8.48 ± 0.86 mM; afterward, CAT declined sharply along with a minor MA generation, ultimately resulting in CAT depletion and MA accumulation (1.60 ± 0.19 mM) at 60 h (Fig. 5b). Here, MA was synthesized because Pp-TC contains the secondary catechol

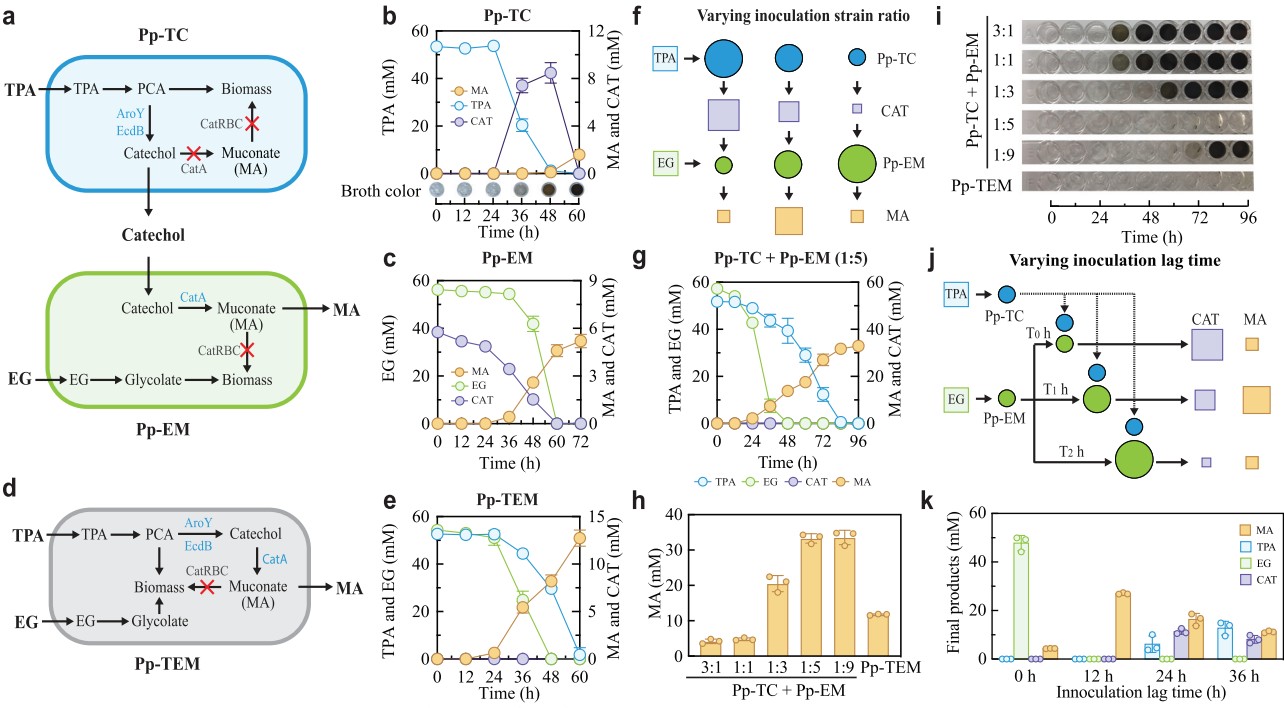

**Fig. 5 | MA production via rational pathway engineering and DOL-mediated optimization.** **a** Design of a synthetic consortium that upcycles PET hydrolysate into MA. The consortium (TC-EM) consists of Pp-TC and Pp-EM derived from Pp-T and Pp-E, respectively. Involving two layers of DOL, the strains were divided to specialize in TPA (Pp-TC) and EG (Pp-EM) consumptions and to enable the TPA-to-CAT (Pp-TC) and CAT-to-MA (Pp-EM) conversions. **b** Temporal profiles of TPA fermentation by Pp-TC that yields CAT. **c** Temporal profiles of EG fermentation by Pp-EM that upgrades CAT into MA. **d** Design of a single strain (Pp-TEM) that produces MA from PET hydrolysate. **e** Temporal profiles of mixed TPA and EG fermentation by Pp-TEM that converts TPA and EG into MA. **f** Production optimization by modulating the initial strain inoculation ratio of the TC-EM consortium. **g** Temporal profiles of PET hydrolysate fermentation by the TC-EM consortium with

a 1:5 initial ratio. **h** Comparison of the final MA production by the TC-EM consortium with different initial ratios and Pp-TEM. **i** Colors of the TC-EM co-culture and Pp-TEM monoculture through the course of hydrolysate fermentation. **j** Production optimization by altering the lag time between Pp-TC and Pp-EM inoculations. As illustrated, Pp-TC was inoculated after Pp-EM upon different durations of delay. **k** Comparison of final products and remnant substrates of the hydrolysate fermentation by the TC-EM consortium with varied inoculation lag times. For panels **a** and **d**, blue text indicates overexpressed genes whereas red cross ('×') indicates the deletion of genes in dark gray. Experimental data are presented as mean values with standard deviations from three independent experiments. Source data are provided as a Source Data file.

1,2-dioxygenase (*catA2*), which plays as a safety valve to convert excess CAT to MA[62]. Additionally, the dynamic CAT fluctuation was accompanied by the change of culture color from transparent to brown and black, which was caused by the mixing and reaction of CAT and $Fe^{3+}$ that yields brown carbon[63] (Supplementary Fig. 10). Pp-EM showed a rapid EG consumption without any MA production in the absence of CAT (Supplementary Fig. 11a). When a low to intermediate level (2.48-5.77 mM) of CAT was supplemented at the beginning of the experiment, Pp-EM not only fully consumed EG but also converted CAT to MA (Supplementary Fig. 11b, c). For example, Pp-EM depleted 56.23 mM EG and yielded $5.20 \pm 0.42$ mM MA from 5.77 mM CAT in 60 h, reaching nearly 90% of CAT conversion (Fig. 5c). By contrast, the control Pp-ES, which possesses the native *catRBCA* cluster, consumed EG but did not yield MA with or without CAT (Supplementary Fig. 11e, f), which was because the operon *catRBC* enabled the organism to convert MA from CAT into biomass in the presence of MA. Importantly, there was an increasing delay and brown carbon accumulation in fermentation with the increase in CAT. Notably, Pp-EM was fully inhibited in the presence of 7.59 mM CAT (Supplementary Fig. 11d, g). Surprisingly, when CAT was supplemented after Pp-EM entered its exponential growth phase (e.g., after 12 h), it affected neither cell growth nor EG assimilation and was fully converted to MA with a 100% yield (Supplementary Fig. 12), suggesting that CAT strongly impairs initial cell growth but not the catalysis underlying the bioconversion.

For comparison, we also devised a control strain, Pp-TEM, that consumes both TPA and EG for MA production (Fig. 5d). The strain was created by transforming the plasmids, which contain the TPA degradation and CAT biosynthesis pathways used in Pp-TC, into Pp-EM. Confirming the design, a mixed TPA and EG fermentation (56.2 mM each) by Pp-TEM resulted in complete substrate utilization and the synthesis of $12.74 \pm 0.85$ mM MA (Fig. 5e).

Compared with Pp-TEM, the strains Pp-TC and Pp-EM were each assigned with reduced tasks regarding TPA and EG catabolism, MA synthesis, CAT detoxification, and stress response. Additionally, by separating the MA pathway at the CAT node, the TC-EM consortium possesses a unique degree of freedom for production optimization. Specifically, by varying the initial Pp-TC:Pp-EM ratio, the CAT influx from TPA and outflux to MA can be altered, thereby enabling one to balance metabolic fluxes and minimize CAT accumulation (Fig. 5f). Indeed, hydrolyzed PET fermentations with different inoculation ratios yielded varied population dynamics, temporal metabolic patterns and final titers that are intercorrelated with each other (Fig. 5g, h and Supplementary Fig. 13). At an initial ratio of 3:1, 1:1 or 1:3 (Supplementary Fig. 13a–f), Pp-TC and Pp-EM grew exponentially through TPA and EG consumption in the beginning phase of fermentation. Concurrently, CAT accumulation was observed, suggesting that the CAT synthesis by Pp-TC was greater than the CAT-to-MA conversion by Pp-EM and that cells faced an increasing level of toxicity from CAT. Over time, cellular growths gradually decreased and eventually, the populations dropped with the population turning points varying from 24 to 48 h. In these cases, EG consumption was incomplete, MA production was inefficient, and there was the accumulation of brown carbon as indicated by the darkening of the culture (Fig. 5i and Supplementary Fig. 13m). At the optimal 1:5 ratio (Fig. 5g and Supplementary Fig. 13g, h), Pp-TC and Pp-EM both managed to grow up to their steady states and maintain their populations, yielding a stable community composition over time. Throughout the process, TPA and EG were fully consumed without CAT accumulation and culture blackening (Fig. 5g, i and Supplementary Fig. 13m). Accordingly, the fermentation yielded a total of $33.28 \pm 1.34$ mM ($4.73 \pm 0.19$ g/L) final MA that was 2.84 fold higher than that from the Pp-TEM fermentation (Fig. 5h and Supplementary Fig. 13k, l). Notably, although a *P. putida* KT2440 strain was reported to have a comparable MA titer (4.59 g/L) from TPA, it required glucose as an additional carbon source and was

significantly impaired in terms of EG metabolism (Supplementary Table 4)[52]. At the 1:9 ratio (Supplementary Fig. 13i, j), Pp-TC and Pp-EM populated simultaneously along with substrate assimilation. However, due to its high initial abundance, Pp-EM depleted its target substrate (i.e., EG) much earlier than Pp-TE which assimilated TPA, and its population started to decay before Pp-TE reached its maximal population at 72 h. The mismatch of these population dynamics was associated with the detection of CAT and blackening of the culture at the end of fermentation, which arose from the reduction of the CAT-to-MA conversion because of the decline of the Pp-EM population (Supplementary Fig. 13i, j and Fig. 5i). These results unraveled the intricate correlation among ecosystem composition dynamics, metabolic patterns, and fermentation outcomes, and also demonstrated that altering initial inoculation composition is an effective strategy to optimize the MA biosynthesis.

In principle, MA production can be additionally modulated by varying the time lag between Pp-TC and Pp-EM inoculations, owing to the facts that efficient MA production requires rapid CAT-to-MA conversion to minimize the CAT accumulation and pre-inoculating Pp-EM enhances the catalysis of the bioprocessing (Fig. 5j). We thus performed another set of hydrolysate fermentations, with Pp-EM being inoculated at the beginning but Pp-TC supplemented later at different time points. Among four conditions tested (Supplementary Fig. 14a–h), we found that a delay of 12 h gave rise to the optimal performance with the completed TPA and EG consumption, no CAT accumulation, and highest MA production (Supplementary Fig. 14c, Fig. 5k). Meanwhile, the two strains, Pp-TC and Pp-EM, both grew up to and maintain their steady-state populations throughout the course of fermentation (Supplementary Fig. 14d). The concurrence of the stable population dynamics and the optimal fermentation performance underscored the intercorrelation between the composition dynamics and metabolic functioning of the consortium. By contrast, no or excessive delays (0, 24, or 36 h) led to incomplete or delayed substrate utilization and CAT accumulation, which were associated with population crashing (the 0 h case) or decline (the 24 and 36 h cases) (Supplementary Fig. 14a, b, e–h). Consistent with these findings, the cultures turned dark (Supplementary Fig. 14i, j) and the MA production was suboptimal (Fig. 5k). These results established inoculation time delay as an alternative to optimizing the ecosystem dynamics and MA production. Remarkably, the compositional patterns, metabolic characteristics, and fermentation outcomes of the consortium under the optimal time lag (12 h) (Supplementary Fig. 14c, d) were highly similar to those at the optimal inoculation ratio (1:5) during the compositional modulation (Supplementary Fig. 13g, h), suggesting that the two parallel strategies converged to a single optimal solution for MA production. Collectively, our experiments demonstrated that, in addition to realizing complete consumption of the PET hydrolysate, microbial DOL provides a powerful tunability for optimizing chemical synthesis.

## Discussion

Through the engineering of microbial DOL, we established a synthetic biology platform that degrades PET hydrolysate and subsequently generates desired products of interest. Compared with its mono-culture counterpart, in our experiments, the consortium was more effective in co-assimilating TPA and EG, particularly when their concentrations were high or crude hydrolysate was directly used. It also surpassed single strains in terms of upcycling performance for both mcl-PHA and MA biosynthesis, by conferring flexible system optimization and reducing catabolic crosstalk and chemical toxicity. These results advance the conceptual understanding of synthetic ecosystems by systematically comparing rationally engineered consortia with monocultures regarding substrate co-utilization, product synthesis, and population-based system optimization. The findings also mirror the prevalence of multispecies microbial communities in native contexts including the settings of plastic biodegradation. In fact, recent

findings showed that plastic debris in environments is often associated with multispecies ecosystems that are considered beneficial for plastic deconstruction[64,65].

In parallel, this study provides valuable insights into the practices of plastic upcycling. As demonstrated, simultaneous and complete TPA and EG consumption through DOL allowed the microbial consortium to reduce substrate assimilation time and to fully upcycle PET hydrolysate. It also avoided EG accumulation which increasingly inhibits cell metabolism over time, therefore opening a possibility for robust fermentation via a fed-batch or continuous fashion. The production of mcl-PHA and MA further demonstrated that hydrolyzed PET alone is sufficient to serve as a sole carbon source for product synthesis with a promising level of titer and yield (Supplementary Tables 3 and 4). Moreover, exemplified via the fine-tuning of MA production, altering inoculation composition and time has been proven as a simple but effective means to optimize fermentation performance. Although engineered microbial consortia will benefit from additional developments such as further minimization of metabolic crosstalk, the highlighted traits conferred by DOL aid in boosting the economic viability of biological PET upcycling for large-scale use.

Beyond their utilities for mcl-PHA and MA synthesis, our designer upcycling consortia can be adapted to produce other chemicals by introducing new biosynthetic pathways and connecting them to the primary metabolic nodes in TPA and EG catabolism by harnessing their modular architecture. Additionally, while the study was limited to PET upcycling, the underlying concept and strategies are potentially applicable to the treatment of other types of plastics, thereby offering insight into the development of a sustainable bioeconomy.

## Methods

### Strains, plasmids, and culture condition

All bacterial strains and plasmids used or constructed in this study are listed in Supplementary Table 1. Primers used in this work for DNA fragment amplification or verification are listed in Supplementary Table 2. Routine cultures of *Pseudomonas putida* and *Escherichia coli* for plasmid construction and preculture were grown in 5 mL Luria Broth (LB) medium with appropriate antibiotics (50 µg/mL kanamycin or 100 µg/mL streptomycin) and at 30 and 37 °C, respectively. All batch fermentations were conducted using 50 mL modified M9 minimal medium supplemented with specific carbon sources at 30 °C and 250 rpm. To address the poor solubility of the free acid form of TPA in water, $Na_2TPA$ was used as a substrate, along with EG, for pure TPA/EG fermentation.

### Plasmid and strain construction

All synthetic oligonucleotides and DNA fragments were ordered from Integrated DNA Technologies (IDT, Coralville, IA). Polymerase chain reaction (PCR) was performed using Q5® High-Fidelity DNA Polymerase (NEB, Lpswich, MA). DNA sequencing services were provided by GENEWIZ from Azenta Life Sciences (Chelmsford, MA). Plasmids were assembled via Gibson Assembly and directly transformed into relative *P. putida* strains. For gene integration, plasmids were constructed using the vector pK18mobsacB and transformed into *E. coli* DH10β competent cells for plasmid amplification. All of the plasmids constructed in this study were verified using restriction digestion and PCR sequencing.

To construct the Pp-T strain, the *ped* operon in *P. putida* EM42 was first knocked out using the plasmid pK18-ped to derive the strain Pp01. Here, pK18-ped was created by cloning the upstream and downstream homology regions of the *ped* operon from *P. putida* EM42 with the primers ped-UF/UR and ped-DF/DR into the vector pK18mobsacB. Pp01 was then confirmed by colony PCR with the primers V-ped-F/R. Meanwhile, after amplification with the primers Ptac-TPA-F/R and TPA-F/R, the tac promoter ($P_{tac}$) and *tpa* operon, including *tpaAa*, *tpaAb*, *tpaC*, *tpaB*, and *tpaK*, from *Rhodococcus jostii* RHA1 were ligated into

the pBb(B5)1k-GFPuv backbone using Gibson assembly to generate the plasmid pBb(B5)1k-tpa. Finally, pBb(B5)1k-tpa was transformed into *P. putida* EM42 and Pp01 to yield the strains Pp-$T_0$ and Pp-T for TPA degradation.

The Pp-E strain was created by first deleting *gclR*, the repressor gene of the Gcl pathway, from the *P. putida* M31 chromosome using the plasmid pK18-gclR. Here, pK18-glcR was constructed by amplifying the upstream and downstream regions of the *gclR* gene by PCR with the primers gclR-UF/UR and gclR-DF/DR and inserting them into the vector pK18mobsacB. After counter-selection with sucrose, the loss of the *gclR* gene was confirmed with colony PCR using the primes V-gclR-F/R. The resulting strain was named PpM01. Similarly, to replace the native promoter of *glcDEF* with the tac promoter, the upstream and downstream homologous regions of the native promoter were amplified by PCR with the primers glcDEF-UF/UR and glcDEF-DF/DR and inserted into the vector pK18mobsacB to derive the plasmid pK18-Pro-glcDEF. Subsequently, the native *glcDEF* promoter of PpM01 was replaced by the tac promoter using pK18-Pro-glcDEF, resulting in the strain Pp-E for EG degradation.

The Pp-TE strain for TPA and EG co-degradation was obtained by transforming the plasmid pBb(B5)1k-tpa into Pp-E.

### Strain construction for mcl-PHA production.

To make the plasmid pSEVA421-phaG-alkK-phaC1-phaC2, the pSEVA421 backbone with the promoter $P_{EM7}$ was first amplified with EM7-F/R. Then, the genes *phaG*, *alkK*, *phaC1*, and *phaC2* were amplified from the genomic DNA of *P. putida* EM42 using the following primers: phaG-F/R, alkK-F/R, phaC1-F/R, and phaC2-F/R, respectively. These five PCR products were circularized by Gibson assembly to obtain the plasmid pSEVA421-phaG-alkK-phaC1-phaC2, which was transformed via electroporation into Pp-T, Pp-E, and Pp-TE to derive the strains Pp-TP, Pp-EP, and Pp-TEP, respectively. On the other hand, genome editing was performed as follows. The upstream and downstream homologous regions (800-1,100 bp) of *fadBA*, *fadBAxE*, and *phaZ* were amplified using the primers fadBA-UF/UR, fadBA-DF/DR, fadBAxE-UF/UR, fadBAxE-DF/DR, phaZ-UF/UR, and phaZ-DF/DR, respectively. Each pair of the corresponding homologous regions was combined with the pK18mobSacB backbone to create the plasmids pK18-fadBA, pK18-fadBAxE, and pK18-phaZ. These plasmids were subsequently used for the deletion of *fadBA*, *fadBAxE*, and *phaZ* in Pp01 and Pp-E. The resulting strains, Pp05 and PpΔ-E, were confirmed by colony PCR with the primers V-fadBA-F/R, V-fadBAxE-F/R, and V-phaZ-F/R. Later, pSEVA421-phaG-alkK-phaC1-phaC2 was transformed into PpΔ-E, yielding the strain PpΔ-EP. Meanwhile, pBb(B5)1k-tpa and pSEVA421-phaG-alkK-phaC1-phaC2 were both transformed into Pp05 and PpΔ-E to generate the strains PpΔ-TP and PpΔ-TEP. To construct control strains, the plasmid pSEVA421 was transformed into Pp-T, Pp-E, and Pp-TE, resulting in the strains Pp-TS, Pp-ES, and Pp-TES, respectively.

### Strain construction for MA production.

The plasmid pK18-catRBCA was constructed by cloning the upstream and downstream homology regions of the *catRBCA* cluster from *P. putida* EM42 with the primers catRBCA-UF/UR and catRBCA-DF/DR into the vector pK18mobsacB. The plasmid was then used to knock out the entire *catRBCA* cluster in Pp01 to eliminate MA utilization. The resulting strain, Pp07, was confirmed by colony PCR with the primers V-catRBCA-F/R. To construct the plasmid pSEVA421-aroY-ecdB, codon-optimized *aroY* and *ecdB* from *Enterobacter cloacae* ATCC 13047 were synthesized by IDT and assembled into pSEVA421 that contains the tac promoter. Finally, pSEVA421-aroY-ecdB and pBb(B5)1k-tpa were both transformed into Pp07 to yield the strain Pp-TC. To integrate the tac promoter upstream of *catA* and delete *catRBC*, the 5' 3' homology regions of *catRBC* were amplified from PpM01 with the primers catRBC-UF/UR and catRBC-DF/DR. The tac promoter was amplified from pBb(B5)1k-tpa using the primers catRBC-ptac-F/R. Then, these fragments were

assembled into the vector pK18mobsacB, resulting in the plasmid pK18-catRBC. Afterward, pK18-catRBC was transformed into PpM02 to yield PpM09. Finally, pSEVA421 and pBb(B5)1k-tpa were further transformed along with pSEVA421-aroY-ecdB into PpM09, generating the strains Pp-EM and Pp-TEM, respectively.

## Transformation of constructs into *P. putida* strains

Electroporation was used for plasmid transformation as described previously[41] with modifications. After overnight cultivation in 5 mL LB medium at 30 °C and 250 rpm, 1 mL *P. putida* cells were centrifuged and washed twice with 1 mL of 0.3 M sucrose solution. Then, cell pellets were resuspended in 0.2 mL of 0.3 M sucrose solution. Plasmid DNAs (0.2–2 µg) were mixed with 100 µL cell suspension in 0.1 cm electroporation cuvette (Bio-Rad) and electroporated at 1.6 kV by Eporator (Eppendorf, Germany). Subsequently, cells were transferred to 1 mL LB medium for recovery at 30 °C and 250 rpm for 1–2 h. 100 µL recovered cells were plated on LB agar plates containing appropriate antibiotics and incubated at 30 °C overnight. Colonies were picked up from agar plates to selective LB medium and grown at 30 °C and 250 rpm overnight for further verification by target fragment PCR and plasmid digestion. For sucrose counter-selection, colonies selected on LB plates were incubated at 30 °C and 250 rpm for 4–6 h. Then, 100 µL cells were plated on LB + 10% sucrose agar plates overnight. Finally, 20–40 colonies were selected to screen for correct gene replacements or knock-out by PCR.

## Characterization of TPA and EG degradation

Engineered *P. putida* strains from glycerol stocks were grown overnight in 5 mL LB medium containing appropriate antibiotics at 30 °C. 2% (v/v) of cells were further inoculated into 50 mL LB medium as second seed culture and cultivated at 30 °C and 250 rpm for 10–12 h. For monoculture cultivation, precultures were then washed twice with distilled water and inoculated at $OD_{600}$ of 0.2 in 50 mL modified M9 minimal medium with TPA and/or EG as the carbon sources at 30 °C and 250 rpm. For co-culture cultivation, cell centrifugation, washing, and resuspension were performed as for monoculture. The initial $OD_{600}$ of the co-culture after inoculation was controlled at 0.2 and the Pp-T:Pp-E ratio was fixed at 1:1 or 3:1 unless otherwise noticed. Fermentation samples (1 mL) were collected at 12 h (or shorter when TPA/EG were consumed mostly) intervals to determine cell growth and TPA and EG concentrations.

## Determination of community composition dynamics

Colony forming unit (CFU) counting was used to analyze the population composition of the consortia. For each co-culture, a 100 µL sample was diluted between $10^4$ and $10^6$ times and spread onto two types of LB agar plates with corresponding antibiotics. After 24 h of incubation, CFUs were counted to determine the growth of each strain as well as the ratio of the strains in a sample. In the case of the T-E consortium, the population of the Pp-T strain carrying the pBb(B5)1k-tpa plasmid was determined by counting CFUs on kanamycin LB agar plates, while the population of Pp-E was obtained by subtracting the number of Pp-T colonies from the total colonies on kanamycin-free LB agar plates. Similarly, for the TC-EM consortium, the Pp-TC population was measured by counting CFU on LB agar plates with kanamycin and streptomycin, whereas the Pp-EM population was calculated by subtracting the number of Pp-TC colonies from the number of total colonies on streptomycin LB agar plates. Control experiments were carried out to count the colonies of Pp-TE and Pp-TEM on LB agar plates supplemented with kanamycin and kanamycin/streptomycin antibiotics, respectively.

## Depolymerization of PET via alkaline hydrolysis

PET powder (ES30-PD-000131) purchased from Goodfellow was used for depolymerization. Briefly, PET powder with 1–20% aqueous NaOH in 500 mL glass bottle was autoclaved for depolymerization in an autoclave reactor at 130 °C for 2 h. After cooling down, the solution was analyzed in terms of its TPA and EG concentrations via a high-performance liquid chromatography system (HPLC). The highest depolymerization reaction was reached at a 5% NaOH solution mixed with benzalkonium chloride (BKC) as a phase transfer catalyst, which was chosen for the following experiments. Batch fermentations with depolymerized PET as the substrate were studied in 250 mL shaker flasks, with each containing 50 mL of modified M9 medium. The depolymerized PET solution with M9 medium was neutralized with HCl and the additional volume of the solution was based on the required TPA and EG concentrations.

## mcl-PHA characterization and quantification

For mcl-PHA production, precultures were prepared as described above and inoculated at an $OD_{600}$ of 0.2-0.4 in modified M9 minimal medium. Nitrogen-limiting condition was prepared by substituting $NH_4Cl$ from 1 to 0.6 g/L in M9 medium. For fed-batch fermentation, pulse addition of feed media (PET hydrolysate and M9 salts) was conducted when TPA/EG were consumed mostly. 40 mL of fermentation broth were taken when TPA and/or EG were totally consumed, and centrifuged at $10,000 \times g$ for 10 min. The cell pellets were then washed with distilled water and lyophilized for PHA extraction. For mcl-PHA composition analysis, 30 mg of lyophilized cells were suspended in 2 mL of sulfuric acid-methanol solution (15%, v/v) and 2 mL of chloroform. The mixture was sealed and placed in a heating block at 100 °C for 4 h. After cooling to room temperature, 1 mL purified water was added to form a two-phase liquid. Then, the organic phase liquid containing the resulting methyl esters was transferred into GC vials and analyzed by gas chromatography–mass spectroscopy (GC–MS) using a SHIMADZU GC2010 equipped with GCMS-QP2010 Plus (Tokyo, Japan). Here, the separation was carried out with a 30-m long column (J&W DB-5ms, film thickness: 0.25 µm; diameter: 0.25 mm) from Agilent (Santa Clara, CA). The GC–MS method was used as previously described with minor modifications[50]. Helium was used as the carrier gas with a flow rate of 1 mL/min. The injection, ion source, and interface temperature were maintained at 250 °C, and the mass spectra were recorded under the scan mode in the range of $35–550 m/z$. The column oven was held at 35 °C for 5 min, heated to 225 °C at 15 °C/min, held at 225 °C for 2 min, then ramped again to 250 °C at 15 °C/min, and finally held at 250 °C for 5 min. Shimadzu GCMSsolution and Mass Spectral Libraries and Databases were used to collect and analyze the GC–MS data. All chemicals for mcl-PHA standards, including Methyl 3-hydroxyhexanoate (6C), 3-hydroxyoctanoic acid (8C), 3-hydroxydecanoic acid (10C), 3-hydroxydodecanoate (12C), and 3-hydroxytetradecanoate (14C), were purchased from Sigma Aldrich.

## Characterization of labor partition for MA production

Starter cultures were prepared as described above and inoculated in 50 mL modified M9 medium, supplemented with either 56.2 mM TPA/EG or PET hydrolysate (final TPA/EG is around 50 mM), at 30 °C and 250 rpm. To vary the initial population ratio, the Pp-TC and Pp-EM monocultures were washed twice, and their cell densities were measured. Then, the two cultures were mixed at different ratios (3:1, 1:1, 1:3, 1:5, and 1:9) with a starting total $OD_{600}$ of 0.2. To vary the inoculation lag time, the Pp-EM monoculture was first washed twice with distilled water and inoculated at $OD_{600}$ of 0.1 in 50 mL selected M9 minimal medium at 30 °C and 250 rpm. After 0, 12, 24, or 36 h, precultured Pp-TC was collected, washed, and inoculated at $OD_{600}$ of 0.1 into the Pp-EM culture. Fermentation samples (1 mL) were collected at 12 h intervals to determine TPA, EG, CAT, and MA concentrations.

## Aromatics and MA quantification

TPA, CAT, and *cis-cis* MA samples were collected after culture centrifugation at $15,000 \times g$ for 10 min. Then, supernatants were diluted

50 times before being injected in the HPLC (Shimadzu, Japan) equipped with a UV detector. The related metabolites were separated on a C18 LC column (Phenomenex Luna 5 μm C18 (2), 100 Å, 150 × 4.6 mm, Torrance, CA) at 40 °C and a flow rate of 0.5 mL/min. The following gradients of buffer A (0.005% Formic acid) and buffer B (20% Acetonitrile) were used: buffer B was first increased from 60% to 80% for 2 min, then ramped to 100% for 6 min, and held at 100% for 12 min. TPA, CAT, and MA were detected at 260 nm with the quantifications being realized using external standards. Shimadzu LCsolution was used to collect the HPLC data. The standard curves were generated by measuring 0.001–0.1 g/L disodium terephthalate (99.0 + %, TCI America™), 0.05–0.5 g/L CAT (≥99.0%, Sigma Aldrich), and 0.001–0.1 g/L MA ((≥97.0%, Sigma Aldrich), respectively.

## EG, glycolate, glyoxylate, and succinate quantification

Samples were first centrifuged at 15,000×$g$ for 10 min, and then the supernatants were diluted 10 times and determined by HPLC (Agilent Technologies 1260 Infinity system, Santa Clara, CA) equipped with a refractive index detector (RID) and a Rezex™ ROA-Organic Acid H+ (8%) LC column (Phenomenex Inc., Torrance, CA). The column was eluted with 5 mM of sulfuric acid as a mobile phase at a flow rate of 0.6 mL/min and 50 °C. Agilent OpenLab ChemStation C.01.10 was used to collect the HPLC data. The pure EG, glycolate, glyoxylate, and succinate purchased from Thermo Fisher Scientific (Waltham, MA) were used to establish the calibration curve for EG, glycolate, glyoxylate, and succinate concentration calculation.

## Statistical analysis

All experiments were performed multiple times as mentioned in figure legends. The results are presented as mean values with standard deviations unless stated otherwise. All statistical analysis was performed using GraphPad Prism V9.1.1. One-way analysis of variance (ANOVA) with Dunnett's multiple comparison tests was used for the comparison of more than two groups. $p < 0.05$ was considered statistically significant.

## Reporting summary

Further information on research design is available in the Nature Portfolio Reporting Summary linked to this article.

## Data availability

Sequences for promoters and genes are provided in Supplementary Information. Other data generated in this study are provided in the Source Data file. Source data are provided with this paper.

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

## Acknowledgements

The authors would like to thank Dr. Víctor de Lorenzo and the Standard European Vector Architecture (SEVA) platform for kindly sharing materials. This work was supported by the Defense Advanced Research Projects Agency via the ReSource program cooperative agreement HR00112020033 (T.L.) and the Future Insight Prize sponsored by Merck KGaA (Darmstadt, Germany) (T.L.). The views, opinions, and/or findings expressed are those of the authors and should not be interpreted as representing the official views or policies of the Department of Defense or the US Government.

## Author contributions

T.L. conceived the project; T.B., Y.C., Y.X., J.J.C., and T.L. designed the study; T.B., Y.C., and Y.X. performed the experiments and collected the data; T.B., Y.C., Y.X., and T.L. analyzed the data; T.L., J.J.C., and T.B. wrote the paper with the input from all authors.

## Competing interests

The authors declare the following competing interests. Patent applicant: The Board of Trustees of the University of Illinois. Name of inventor(s): T.L., T.B., Y.Q., and Y.X. Title: Engineered Microbial Consortia for

Upcycling Polyethylene Terephthalate. Application number: 63/509,709. Status of application: Pending. Specific aspect of manuscript covered in the patent application: Systems and methods for the degradation and upcycling of polyethylene terephthalate. All other authors declare no competing interests.
