## [Peer Review File · Nature Communications]

Reviewers' Comments:

Reviewer #1:

Remarks to the Author:

In order to upcycle plastics into value-added products, the authors engineered a *Pseudomonas putida* synthetic microbial consortium to degrade polyethylene terephthalate (PET) hydrolysate. First, they engineered and characterized two strains each specializing in the consumption of one of the two main PET hydrolysate products, terephthalic acid (TPA) and ethylene glycol (EG), respectively. They showed that each strain is specific in the substrate it utilizes, and that the consortium efficiently degrades both substrates when co-cultured. They also characterized how robustly the cells ferment the substrate they specialize in, in the presence of various concentrations of the other substrate and found that the TPA utilization strain is less sensitive to increasing EG concentrations. Next, they constructed a single strain capable of utilizing both TPA and EG, and compared its performance with their synthetic consortium. They found that the consortium consistently degrades the mixture faster than its single strain counterpart, especially when one of the two substrates was abundant. Moreover, they observed that the consortium consumes both substrates simultaneously whereas the single strain does so sequentially, which they attribute to the alleviation of metabolic crosstalk between the two pathways.

The authors then proceeded to establish a pipeline for PET hydrolysis and used the hydrolysate as a substrate. They show that the consortium still outperforms the single strain, however both are unable to process high hydrolysate concentrations, which they could in the pure mixture case, pointing towards additional hydrolysate toxicity. To achieve PET upcycling they further engineered each of their specialized strains, as well as the co-utilization strain to produce medium chain length polyhydroxyalkanoate (mcl-PHA). Two variants were engineered and characterized for each strain and the best performing ones were selected to be compared in a consortium vs single strain experiment. The consortium achieved complete hydrolysate degradation, whereas the single strain was unable to fully utilize EG, as well as a higher mcl-PHA titer. The authors then asked whether the division of labor they engineered could alleviate toxicities caused by the accumulation of intermediate metabolites during muconate (MA) production. To address this, the MA production pathway was split between the TPA and EG utilizing strains and a control strain was built by introducing the full pathway in the co-utilization strain. Each strain was characterized first in a monoculture and then the modularity of the consortium was leveraged to optimize MA production by tuning the initial strain ratio. The optimized consortium achieved significantly higher MA titer and showed no intermediate metabolite accumulation. Finally, the modularity of the consortium was leveraged again to demonstrate simultaneous production of two distinct products from a single fermentation. The two specialized strains were engineered for simultaneous mcl-PHA and itaconic acid production. Simultaneous production was demonstrated both in monoculture as well as in the consortium.

Overall I think this is good work, possessing strong experimental design and a compelling presentation, and would be of great interest to many in the fields of metabolic engineering, bioremediation and synthetic biology. However, there are some issues I would like to see addressed or clarified in the revised manuscript:

- Was mcl-PHA production the main reason for choosing *Pseudomonas putida* as the chassis organism for this work? Please consider adding a short justification for chassis selection in the introduction.
- Consider rephrasing the discussion of Sup. Fig. 2 (page 5, line 27-29). Both Pp-T and Pp-T0 seem to be sensitive to EG concentration, though Pp-T less so.
- The authors claim that lower metabolic burden might be responsible for the higher degradation efficiency of the consortium compared to the generalist strain in Fig. 2 (end of page 7). In the context of the performed experiments quantifying burden is not straightforward. For example, the generalist strain seems to be equally or more fit than the EG specialist when both are grown with EG as a sole carbon source (Sup. Fig. 3b and 4b). Also, the consortium seems to have a higher maximal growth rate but fails to reach the biomass accumulation of the generalist strain.
- Relevant to the aforementioned, I believe the paper would greatly benefit from a closer look into the growth dynamics of the two strains in the consortium, quantifying the growth of each specialist strain. Now there is only a very indirect measure for each strain, substrate degradation. However, as is evident in Fig. 2d, there is not a good correlation between growth and substrate degradation

(36 - 48hs). Also, since tuning initial strain ratio is used later on to optimize the consortium's performance, it would be invaluable to see how different initial ratios evolve over time.

- There seems to be an additional data point in Fig. 2d, unless this refers to two samples taken very close to each other.
- When discussing Fig. 4g the authors claim that the reduced EG fermentation of Pp-TEP is due to the hydrolysate and compare with data from Sup. Fig. 7 where Pp-TEP is grown on a pure mixture of TPA and EG (page 10, last paragraph). However it is unclear if the data from Sup. Fig. 7 were collected from an experiment performed in nitrogen-limited conditions. If so, please indicate in the figure legend, otherwise a control is needed to determine whether the hydrolysate or nitrogen starvation is responsible.
- Data in Fig. 4k are never elaborated on. Is the improved yield in batch fed fermentation due to more efficient substrate utilization for products instead of biomass? Have any of the relevant enzymes been shown to work better in stationary phase?
- Page 12 line 22: I believe the authors are referring to Fig. 5c instead of Fig. 5d.
- When discussing Fig.5 (page 13, paragraph 2) the authors claim that the two specialist strains have a reduced burden compared to the generalist. There is insufficient data for this claim, both because of the difficulty to quantify burden in this experimental setting and also because there is no growth data for strain Pp-TC.
- The authors demonstrate in Fig.5 and Sup. Fig. 11 that they can optimize their consortium's performance by tuning the initial strain ratio. It would be very interesting to see how this strain ratio evolves as the fermentation proceeds. Also, the data from Fig. 5g (the optimized ratio) show similar behavior to the data from the optimized delay between inoculations (Sup. Fig. 12b) and would be interesting to further probe this by examining the co-culture composition in more detail.
- There seem to be several typos in Supplementary Fig. 9 where the concentrations reported in the main text do not match those reported in the figure (e.g. 5.77mM in the text vs 5mM in the figure).
- Why did the authors choose to engineer both strains to simultaneously produce mcl-PHA and itaconic acid (IA) instead of one strain producing mcl-PHA and the other IA? It seems contrary to the division of labor approach that they follow throughout the paper.
- There is no single strain control to compare the performance of the engineered consortium for mcl-PHA and IA co-production. Why is that? Demonstrating improved performance of the consortium over a single strain generalist is the main point of this work, however, in this case, a comparison cannot be made.
- For Fig.6b,c consider explaining in the legend what event the time points correspond to (as in Fig 4 c-e).
- The authors have chosen to display their time point data as large discs with the error bars extending beyond them when they're larger. This choice obscures the observed variability of the data. This is especially problematic in Sup. Fig. 1-4 where there is significant overlap between data points.

Reviewer #2:

Remarks to the Author:

Engineering Microbial Division of Labor for Plastic Upcycling_404734

The study by Teng Bao and colleagues investigates the possibility of designing synthetic microbial consortia for the upcycling of PET. The authors compare numerous engineered strains of *Pseudomonas putida* for their ability to utilize PET monomers (EG and TPA) and convert them into new high-value bioplastics (mcl-PHAs) and/or building blocks (i.e. muconic acid and itaconic acid). Furthermore, co-cultured strains with different metabolic pathways are compared with the performance of single strains possessing both pathways, to assess the metabolic advantage derived from the division of labor. High concentrations of EG and TPA are assimilated. The key findings are interesting and suggest that the division of labor allows to exploit the substrate in a more efficient way, also reducing eventual inhibiting effects of toxic intermediates. However, the relevance of this technology in terms of plastic upcycling should be discussed in a more thorough way, also comparing the results with the state of the art.

Overall comment: this is a very extensive and interesting work but it feels somewhat of a

patchwork of studies, rather than a coherent story. It might be therefore beneficial to highlight better the connection between the different sections. The authors might actually consider if presenting all the results in this manuscript, based on what are the key findings they want to convey. Sometime, "less can be more". Explaining better what is the real scientific question the authors want to investigate, what mechanism to be unveiled, and how the experimental set up is expected to answer these question could for instance also help creating a link between the different parts. A clearer focus would help the reader and could avoid giving the impression of different upcycling studies put together.

More in detail:

The authors present a remarkable amount of experiments and a robust experimental investigation with comparison of over 16 different substrate combinations and a significant amount of mutants. However, some important aspects should be carefully addresses before publication.

- In several cases there is an unproper use of technical terms regarding biodegradation of polymers and concepts regarding upcycling of plastics. The authors write for instance about the ability of their strains to perform "plastic deconstruction" in tests in which they were actually were using TPA and EG. In the abstract the authors describe the "degradation" of PET hydrolysate, and there is a confusion between the goal, which is to upcycle plastics but also to "achieve complete degradation". In another paragraph they talk about "plastic deconstruction" while referring to the degradation of TPA and EG. In yet other parts the term "biodegradation" seems to be confused with "assimilation". A more careful use of these terms is advisable.
- Some of the statements of the authors show an incomplete knowledge of the most recent articles published in the field of plastic upcycling and high-value building blocks, so I would recommend diving deeper in the most recent papers and literature.
- The introduction is a bit too generic and would benefit of a more precise description of the scientific question and the experimental plan to answer that question: there seem to be multiple tasks (degrade PET, microbial uptake of the monomers, study of division of labor, upcycling to 3 different high-value products, etc.). So a more sharp focus and description of the key scope of the study would be beneficial. I would suggest creating a sort of "red thread" that shows better how the different goals and parts of the study are connected with each other, to provide the impression of a more coherent work.
- In some cases, the conclusions are not completely supported by the experimental data presented, but represent more the starting hypothesis: for instance, in page 7, the authors argument that "by assigning the pathways to separate strains" this leads to the advantage of "carrying a lower metabolic burden". Even though since is what is expected in theory, not all data can be explained in this way and this should be then critically discussed. It could be argued, for instance, that a lower metabolic burden should then lead to more energy and carbon available for biomass production for the synthetic consortium. However, in Fig 2d) and 2e) the co-culture Pp-T + Pp-E shows a lower biomass production (max OD = 12 against 15) and the biomass growth curve shows almost a diauxic growth pattern, which is not expected in the synthetic consortium. I would therefore recommend that this part of data interpretation regarding the advantage of the division of labor should be elaborated a bit more, to provide more inn-depth discussions, supported by the data in a more comprehensive way.
- Experimental plan: In most experiments, pure EG and TPA is used and in some cases the real PET hydrolysate is also used to evaluate the effect of inhibiting compounds. It is very important and highly relevant to use directly the diluted crude product from the hydrolysis, in view of a real bioupcycling process that can be scaled up in the future. On the other hand, the real PET hydrolysate was not tested in all experiments and that might lead to some confusion in the reader. It might also be a good idea to underline that when the authors are writing about "TPA" they actually worked with Na₂TPA (at least when using the hydrolysate), which is known to have a better solubility and lower toxicity to the cells compared to pure TPA. Since the authors managed to work with 100 mM TPA I would guess that was also disodium terephthalate? It might be worth

clarifying, because in several cases I was in doubt whether pure TPA was used (as stated in the text) or rather Na₂-TPA.

- In the paragraph dedicated to the Upcycling of PET hydrolysate into mcl-PHA, I would recommend mentioning also that there are already studies that have used *P. putida* to upcycle PET monomers into mcl-PHAs. Else it seems that this as this is the first study looking into this, while it has been investigated already for 10 years and there are start-up companies from universities already scaling up the technology, using engineered *Pseudomonas* strains.

- To highlight better the relevance of the experimental findings in terms of TPA and EG upcycling into the different high-value compounds (and facilitate comparisons with literature), I would recommend presenting a table with the best results expressed in Cmol and the %Cmol that from the substrate ends up into the final product. The fact that an engineered strain is able to perform a new metabolic pathway is of course a very relevant results from a synthetic biology point of view but its real impact from the plastic upcycling point of view depends also on final titers and conversion efficiencies. Since not all the readers are necessarily experts in plastic bioupcycling, this would help underlining the relevance of your results and will also facilitate the discussion part, where the authors can compare their findings against the literature findings.

- In the paragraph regarding muconic acid biosynthesis via DOL, it seems like the experimental design (of the synthetic consortium) is very different compared to the previous sections, so the rationale behind this choice should be clarified better. I would also suggest clarifying more in detail which experimental results demonstrate crosstalk reduction in your synthetic consortium compared to the single engineered strain. I would also suggest checking more thoroughly the latest literature on this topic: MA biosynthesis has also been investigated using TPA and not only with glucose feeding. In general, a more careful use of literature would be appropriate.

- To sum up, I would say that the authors did a tremendous work from a synthetic biology point of view, while the real impact on the plastic upcycling topic should be highlighted better. The authors could underline more the relevance of their findings, showing how the obtained titers and productivities compare to the requirements for a viable process that can be scaled up in the future (and other studies proposing other bioprocesses for the production of IA or PHA)?

- In section regarding the PHA and itaconic acid production, the rational of this co-production is not very clear to me, as itaconic acid has a market value that can be 10-20 times higher than mcl-PHA. So what would be the point to direct precious carbon to produce PHA instead of maximizing itaconic acid, from a process point of view? This should be clarified so that the reader does not get impression of a mere (high level) synthetic biology "exercise" that, though of high quality, has no real reason or application /impact on the real plastic waste upcycling topic. What is the division of labor advantage in this experiment, in terms of substrate conversion into the 2 products?

- Furthermore, the downstream processing for PHA is a well-known bottleneck exactly because it is intracellular and requires multiple steps to remove from the cells and purify, without changing the thermoplastic properties. On the other hand, IA is secreted and this would be the real simple downstream separation, from a process point of view. So what is the downstream advantage to coproduce PHA when IA could be continuously produced and easily separated?

- On a different note: Please use consistent units for the titer of your products. Presenting PHA in mg/L and IA in mM does not facilitate comparison.

- Also: please explain what you mean with "the titer was slower"? (a titer does not contain a time unit).

- In the Conclusion paragraph, a deeper discussion the meaning of these findings for the division of labor and the relevance for PET upcycling into MA, PHA and IA would be highly relevant, to

better underline the contribution of this study to the advancement beyond the state of the art.

- I am also a bit worried about the soundness of some conclusions, which are not completely supported by experimental data (and if they are, you might consider explaining it more): for instance, I am also not totally clear if the conclusion about the reduced catabolic crosstalk is based on your experimental results (and if yes, which ones exactly?) or rather the initial hypothesis? Furthermore, is the production of 0,6 mM IA relevant enough to be considered as a coproduction (compared to 200 mM of PHA)? 0.6 mM is in any case rather insignificant for the production of a platform chemical.

- So in conclusion, I agree with the great potential of this approach but this needs to be also supported more clearly by the experimental results, which need to be comparable with other biotechnological solutions. So comparison and critical discussion against the state of the art is essential and not sufficiently addressed in this work.

- Regarding the use of literature: Several of the most relevant studies using PET monomers for upcycling to PHA with *Pseudomonas* are missing. Also the comparison with other bioprocesses that upcycle PET into MA is missing, as well as bioprocesses for the production of IA from waste streams. In my view, this is reflected in a not particularly deep discussion about the relevance of the findings of this study.

- Regarding the figures: they are very interesting and of good quality, but in order to fit all the results in a same figure, the readability becomes difficult (too crowded and too small fonts inside the figures). Also in the supplementary material, the concentrations of the substrates in the legend inside the figures is really hard to read, due to the small font size.

- In Fig 2d) and 2e) Pp-T + Pp-E shows a lower biomass production (max OD = 12 against 15) and the biomass growth curve shows almost a diauxic growth pattern, which is not expected in the synthetic consortium. It would be interesting if you could elaborate a bit on this.

- In Fig 4j: It does not help to present some of the data in mM and others as mg/L. I would recommend to uniform this to facilitate the comparison of conversion processes from substrate to final product.

- Fig 4k: the fed batch shows a higher conversion yield? Could you comment on this, also based on literature findings from other studies?

- Also: in your upcycling test with the hydrolyzed PET you use equimolar concentration of TPA and EG. Since you used the raw hydrolysate, does this mean that you obtain an equimolar mixture of TPA and EG from your alkali hydrolysis?

- Figure S7: since these are the results of the PHA producing strains it might make more sense to also show the PHA, else it is not really possible to see obtain the conversion yields

- Fig 9S: could you comment on the difference and the meaning of the results between Fig S9b and 9 f? Why is PP-ES not showing any inhibition to 2.5 mM CAT and converts it at the same rate of EG? Is this conversion growth associated?

- Fig S11: the results of the ratio 1:5 (described as the optimum in the text) are not shown even though the batch is presented in the fig 11f

- Fig S12: the captions should allow to read the figures as a stand-alone, but from the description it is not possible to understand that only one of the 2 strains was added with a delay. I suggest to clarify this point in a few words

REVIEWER COMMENTS

Reviewer #1 (Remarks to the Author):

- Overall I think this is good work, possessing strong experimental design and a compelling presentation, and would be of great interest to many in the fields of metabolic engineering, bioremediation and synthetic biology. However, there are some issues I would like to see addressed or clarified in the revised manuscript.

Author Response: We would like to thank the reviewer for taking time to constructively read our manuscript and provide us valuable comments.

- Was mcl-PHA production the main reason for choosing *Pseudomonas putida* as the chassis organism for this work? Please consider adding a short justification for chassis selection in the introduction.

Author Response: *Pseudomonas putida* was chosen because of its adaptable genetics and versatile metabolic capacity in remediating chemicals and producing high-value products such as mcl-PHA, MA and their derivatives. As suggested, we added a short justification in the beginning paragraph of the Results section as follows: “Here, the soil bacterium *P. putida* was selected as the cellular chassis due to its versatile metabolic capacity, genetic amenability for manipulation, and prior demonstration of valued product production from waste-derived feedstocks.”

- Consider rephrasing the discussion of Sup. Fig. 2 (page 5, line 27-29). Both Pp-T and Pp-T0 seem to be sensitive to EG concentration, though Pp-T less so.

Author Response: As suggested, the relevant discussion in revised paper was rephrased as “We found Pp-T remained efficient in TPA assimilation and biomass accumulation with the increase of EG concentrations for a given TPA level (100 mM) as compared to Pp-T₀ which was much more sensitive to the presence of EG.”

- The authors claim that lower metabolic burden might be responsible for the higher degradation efficiency of the consortium compared to the generalist strain in Fig. 2 (end of page 7). In the context of the performed experiments quantifying burden is not straightforward. For example, the generalist strain seems to be equally or more fit than the EG specialist when both are grown with EG as a sole carbon source (Sup. Fig. 3b and 4b). Also, the consortium seems to have a higher maximal growth rate but fails to reach the biomass accumulation of the generalist strain.

Author Response: We thank the reviewer for this critical comment. As the reviewer mentioned, quantifying the metabolic burden is indeed not straightforward. We acknowledge that it is our speculation from theoretical reasoning that the specialists have a reduced metabolic burden when utilizing TPA and EG as co-substrates, owing to the need to encode two sets of enzymes and associated co-factors for the consumption of TPA and EG. To ensure the rigor and accuracy of the presentation, we have removed the speculation of reduced metabolic burden in our revision.

Meanwhile, relating to the reviewer’s comment on EG consumption, we would like to point out that Supplementary Fig. 3b and 4b did show that the EG specialist (Pp-E) has a slightly faster

rate of substrate utilization than the generalist (Pp-TE) due to a reduced lag time. For the initial conditions of 10, 31.6, 100, 178, and 316 mM, the remaining EG concentrations at hour 12 for Pp-E were 0, 13.98, 84.40, 161.11, and 303.33, whereas those for Pp-TE were 1.62, 20.16, 89.25, 172.23, and 316.74 mM. At hour 36, for the conditions of 178 and 316 mM (10, 31.6 and 100 mM were depleted), the remaining EG levels for Pp-E were 0 and 130.18 mM; while those for Pp-TE were 18.65 and 144.09 mM.

In addition, in concert with the reviewer's comment on biomass production, we were aware of the lower biomass accumulation by the T-E consortium than the generalist Pp-TE. To unravel the possible cause, we repeated the fermentation experiments and analyzed the production of metabolites during the fermentations. Our results (Supplementary Fig. 5b,f) showed that the T-E consortium catabolized TPA and EG more efficiently than Pp-TE but, in the meanwhile, it led to transient glycolate accumulation and subsequent depletion, and also yielded significantly more succinate at the end of fermentation than the generalist Pp-TE. Thus, the results suggested that succinate production likely contributed to the reduction of biomass accumulation in the T-E consortium and redirecting the succinate flux to desired biomass or product synthesis could be a potential target for future investigation and optimization. In the revised paper, we reported these new experimental results and added a discussion about our findings.

- Relevant to the aforementioned, I believe the paper would greatly benefit from a closer look into the growth dynamics of the two strains in the consortium, quantifying the growth of each specialist strain. Now there is only a very indirect measure for each strain, substrate degradation. However, as is evident in Fig. 2d, there is not a good correlation between growth and substrate degradation (36 - 48hs). Also, since tuning initial strain ratio is used later on to optimize the consortium's performance, it would be invaluable to see how different initial ratios evolve over time.

Author Response: We appreciate the reviewer's thoughtful and constructive feedback. As suggested, we conducted colony forming unit (CFU) counting experiments to investigate the temporal composition dynamics of the T-E consortium with varying initial inoculation ratios (5:1, 3:1, 1:1, 1:3, and 1:5). Our results (Supplementary Fig. 5) showed that, associated with the alteration of metabolic patterns, the ecosystem population dynamics and final composition are also subject to the variation of inoculation ratio. Namely, a higher initial Pp-T abundance resulted in a higher final Pp-T population and a higher initial Pp-E abundance led to a higher Pp-E in the final population. We added the experimental results in Supplementary Fig. 5 and discussed the results in the revised manuscript.

- There seems to be an additional data point in Fig. 2d, unless this refers to two samples taken very close to each other.

Author Response: These were two samples taken very close to each other, with the purpose of accurately measuring the time of TPA depletion. Due to the need to repeat the experiment for CFU counting and metabolite analysis for the last comment, we have updated Fig. 2d with newly obtained data.

- When discussing Fig. 4g the authors claim that the reduced EG fermentation of Pp-TEP is due to the hydrolysate and compare with data from Sup. Fig. 7 where Pp-TEP is grown on a pure

mixture of TPA and EG (page 10, last paragraph). However, it is unclear if the data from Sup. Fig. 7 were collected from an experiment performed in nitrogen-limited conditions. If so, please indicate in the figure legend, otherwise a control is needed to determine whether the hydrolysate or nitrogen starvation is responsible.

Author Response: The fermentation kinetics presented in Supplementary Fig. 8 (previously Supplementary Fig. 7) were assessed under pure TPA/EG conditions with sufficient nitrogen supply. Following the reviewer's suggestion, we conducted a hydrolysate fermentation experiment using the TP-EP consortium and the Pp-TEP strain with sufficient nitrogen supplementation (Supplementary Fig. 9). Comparison of the results (Supplementary Fig. 9b) with those in the nitrogen limiting condition (Fig. 4g) showed that, Pp-TEP's EG assimilation was impaired in both cases regardless of nitrogen availability but was worse in nitrogen limitation. Meanwhile, under nitrogen sufficient settings, we noticed Pp-TEP managed to consume both substrates when pure TPA and EG were used (Supplementary Fig. 8h (previously Supplementary Fig. 7h)) but failed to continuously assimilate EG in hydrolysate (Supplementary Fig. 9b), suggesting that the hydrolysate also contributed to the inhibition of EG consumption. Thus, the PET hydrolysate and nitrogen starvation collectively contributed to the reduction of EG consumption. We added these new results to the revised paper.

- Data in Fig. 4k are never elaborated on. Is the improved yield in batch fed fermentation due to more efficient substrate utilization for products instead of biomass? Have any of the relevant enzymes been shown to work better in stationary phase?

Author Response: We apologize for the oversight. Fig. 4k in the original manuscript reported the mcl-PHA production per dry cell weight of the TP-EP consortium and the TEP strain in single batch and fed-batch experiments. We did not measure enzyme activities; instead, we speculate that the slight increase in mcl-PHA production per dry biomass in fed batch were due to two reasons. (1) Reduced biomass increases during the second round of hydrolysate fermentation. Our results showed that, for the fed batch fermentation, the optical density (i.e., OD) of Pp-TEP increased from 0.2 to 8.1 during the initial round of hydrolysate consumption but, upon the assimilation of the second hydrolysate, increased to 14.6. In other words, the Pp-TEP biomass accumulation during the initial and second stages was 7.9 and 6.5 respectively. Similarly, the biomass accumulation of the consortium during the initial and second round was 9.8 and 6.7 respectively. The reduction of biomass increase would lead to a higher yield for the same amount of increase of mcl-PHA. Additionally, reduced biomass production suggested that likely more carbon was directed to the synthesis of other metabolites including mcl-PHA. (2) The second reason was the exaggeration of nitrogen limitation. Due to experimental variations in hydrolysis, we noticed that the actual TPA and EG concentrations in the hydrolysate for the second round of supplementation was a bit higher than those at the first round, which could exaggerate nitrogen limitation that is known to promote mcl-PHA production. These facts collectively led to a higher mcl-PHA production per cell biomass in fed-batch fermentation. Now, we have updated Fig. 4k to present the mcl-PHA yield in terms of substrate weight to be consistent with other panels and moved the PHA production per biomass weight to Supplementary Fig. 9d and discussed the results in the revised manuscript.

- Page 12 line 22: I believe the authors are referring to Fig. 5c instead of Fig. 5d.

Author Response: Yes, it should be “Fig. 5c”. It has been corrected now in the revised paper.

- When discussing Fig.5 (page 13, paragraph 2) the authors claim that the two specialist strains have a reduced burden compared to the generalist. There is insufficient data for this claim, both because of the difficulty to quantify burden in this experimental setting and also because there is no growth data for strain Pp-TC.

Author Response: We thank the reviewer for this comment. We acknowledge that it was theoretical reasoning that the specialists have a reduced metabolic burden owing to the fact that, compared to the generalist, they each need to encode only set of enzymes and associated co-factors for the consumption of TPA or EG. We thus removed the statement about burden reduction from the revised paper and rephrased the statement to ensure the accuracy of the presentation.

- The authors demonstrate in Fig. 5 and Sup. Fig. 11 that they can optimize their consortium’s performance by tuning the initial strain ratio. It would be very interesting to see how this strain ratio evolves as the fermentation proceeds. Also, the data from Fig. 5g (the optimized ratio) show similar behavior to the data from the optimized delay between inoculations (Sup. Fig. 12b) and would be interesting to further probe this by examining the co-culture composition in more detail.

Author Response: We are grateful for the valuable feedback. Following the reviewer’s suggestion, we have repeated the fermentation of the MA-producing engineered consortium under varied initial inoculation ratios and different lags of inoculation time. For the experiments, we conducted CFU counting to reveal the temporal composition dynamics in conjugation with the measurement of substrates, metabolites and products. Our results, presented in Supplementary Fig. 13 and 14 of the revised paper, showed that community composition dynamics is intricately correlated with the temporal metabolic profiles and fermentation outcomes both in the inoculation-ratio case and in the time-delay case. In short, stable growth and population maintenance of the co-culture were associated with complete TPA and EG consumption, absence of CAT accumulation, and maximal MA production; by contrast, population collapse or decay corresponded to incomplete substrate utilization, CAT accumulation and suboptimal MA production. Additionally, the compositional patterns, metabolic characteristics and fermentation outcomes of the fermentation under the optimal time lag (12 h) (Supplementary Fig. 14c,d) were all shown to highly similar to those with the optimal inoculation ratio (1:5) (Supplementary Fig. 13g,h), which suggested that the two optimization strategies converged to a single optimal solution for MA production. Collectively, these experiments uncovered the intricate relationship among population dynamics and ecosystem functioning and demonstrated that microbial DOL offers a powerful tunability for simultaneous optimization of product synthesis and substrate deconstruction.

In the revised paper, we added new data to Supplementary Figs. 13 and 14 (previously Supplementary Figs. 11 and 12) and amended the presentation and interpretation of the experimental results in the corresponding sections of the manuscript (Main Text, highlighted text starting with “Indeed, hydrolyzed PET fermentations ...” (Page 14, the 4th line from the bottom) to the end of the first paragraph in Page 15, and the highlighted text starting with “Among four conditions tested (Supplementary Fig. 14a-h), ...” (The third line of Page 16) to the end of the paragraph).

- There seem to be several typos in Supplementary Fig. 9 where the concentrations reported in the main text do not match those reported in the figure (e.g. 5.77mM in the text vs 5mM in the figure).

Author Response: As suggested, we have corrected Supplementary Fig. 11 (previously Supplementary Fig. 9) in the revised paper.

- Why did the authors choose to engineer both strains to simultaneously produce mcl-PHA and itaconic acid (IA) instead of one strain producing mcl-PHA and the other IA? It seems contrary to the division of labor approach that they follow throughout the paper.

Author Response: We thank the reviewer for this comment. The objective of the experiment was to demonstrate the versatility of the engineered consortia in achieving for complex and advanced bioproduction. To that end, we tested the feasibility of simultaneously producing multiple products, intracellular mcl-PHA and extracellular IA, from hydrolyzed PET in a single fermentation.

[Redacted]

However, we did not include the results due to our concern about the volume of information in the paper, because a complete presentation of the results on the co-production will bring four more figures containing 17 panels. Indeed, consistent with our concern, Reviewer 2 raised the same concern, and explicitly recommended us to remove some of the results in the paper and focus more on elaborating the motivation, context and interpretation of our findings to increase the clarity and coherence of the work. After careful consideration, we decided to follow Reviewer 2's advice of "less can be more" and took the PHA and IA co-production section out of the article. We believe that it helps the manuscript to be more focused and coherent. That being said, we included here our detailed results below as a full response to this comment:

[Redacted]

Finally, we would like to reiterate that, following Reviewer 2's advice, we removed the co-production section from the article to help the manuscript to be more focused and coherent.

- There is no single strain control to compare the performance of the engineered consortium for mcl-PHA and IA co-production. Why is that? Demonstrating improved performance of the consortium over a single strain generalist is the main point of this work, however, in this case, a comparison cannot be made.

Author Response: As detailed in our response to the last comment, we developed the monoculture strain (Pp-TEAPI) and conducted relevant experiments.

[Redacted]

Meanwhile, we want to reiterate that we ultimately followed Reviewer 2's suggestion to remove the co-production results to improve the coherence and readability of the work.

- For Fig. 6b,c consider explaining in the legend what event the time points correspond to (as in Fig 4 c-e).

Author Response: The time points for the data in Fig. 6b,c are shown in the bottom of the panel, and correspond to the sampling time during the PET hydrolysate fermentation. For Fig 4c-e, we added in the legend the corresponding time points.

- The authors have chosen to display their time point data as large discs with the error bars extending beyond them when they're larger. This choice obscures the observed variability of the data. This is especially problematic in Sup. Fig. 1-4 where there is significant overlap between data points.

Author Response: The use of large discs with extended error bars could indeed obscure the variability of the data. Thus, we made two attempts to improve the visualization. One was to place the error bars on top of the discs; the other was to decrease the disc size. However, due to the large numbers of data points presented in these figures, we found that, among the original and the two alternatives, the original figure gave rise to the best presentation of the data for visualization. The other two made it difficult to distinguish discs with different colors. Thus, with all due respect, we decided to keep the original figures.

Reviewer #2 (Remarks to the Author):

- The key findings are interesting and suggest that the division of labor allows to exploit the substrate in a more efficient way, also reducing eventual inhibiting effects of toxic intermediates. However, the relevance of this technology in terms of plastic upcycling should be discussed in a more thorough way, also comparing the results with the state of the art.

Overall comment: this is a very extensive and interesting work but it feels somewhat of a patchwork of studies, rather than a coherent story. It might be therefore beneficial to highlight better the connection between the different sections. The authors might actually consider if presenting all the results in this manuscript, based on what are the key findings they want to convey. Sometime, “less can be more”. Explaining better what is the real scientific question the authors want to investigate, what mechanism to be unveiled, and how the experimental set up is expected to answer these question could for instance also help creating a link between the different parts. A clearer focus would help the reader and could avoid giving the impression of different upcycling studies put together.

Author Response: We thanks the reviewer to taking time to read our manuscript and providing many constructive and insightful comments. We were very glad to hear that the reviewer found our work interesting and extensive.

We also greatly appreciate the overall comment on the improvement of presentation, and fully agree with the reviewer about the “less can be more” philosophy. After careful consideration, we decided to remove the mcl-PHA and IA co-production section from the manuscript. However, we provided the related results and supplementary information at the end of this response letter (Pages 17-20) for the reviewing purpose. Meanwhile, we have significant revised and improved he presentation of the paper by adding additional related background, motivation, result interpretation and connection to elaborate on the results and streamline the manuscript to be more cohesive. Additionally, we amended the manuscript with additional experimental results including the measurement of community composition dynamics to provide more necessary details. We believe that these revisions have addressed the reviewer’s concerns and improved the clarity and coherence of the manuscript.

- In several cases there is an unproper use of technical terms regarding biodegradation of polymers and concepts regarding upcycling of plastics. The authors write for instance about the ability of their strains to perform “plastic deconstruction” in tests in which they were actually were using TPA and EG. In the abstract the authors describe the “degradation” of PET hydrolysate, and there is a confusion between the goal, which is to upcycle plastics but also to “achieve complete degradation”. In another paragraph they talk about “plastic deconstruction” while referring to the degradation of TPA and EG. In yet other parts the term “biodegradation” seems to be confused with “assimilation”. A more careful use of these terms is advisable.

Author Response: We sincerely appreciate the reviewer for careful reading. As suggested, we have corrected the term degradation with assimilation or consumption, and also checked the use of other technical terms and corrected them as necessary.

- Some of the statements of the authors show an incomplete knowledge of the most recent articles published in the field of plastic upcycling and high-value building blocks, so I would

recommend diving deeper in the most recent papers and literature.

Author Response: As suggested, we have conducted a systematic literature review for the most recent papers on plastic upcycling and high-value building blocks, and added related references in proper contexts.

- The introduction is a bit too generic and would benefit of a more precise description of the scientific question and the experimental plan to answer that question: there seem to be multiple tasks (degrade PET, microbial uptake of the monomers, study of division of labor, upcycling to 3 different high-value products, etc.). So a more sharp focus and description of the key scope of the study would be beneficial. I would suggest creating a sort of “red thread” that shows better how the different goals and parts of the study are connected with each other, to provide the impression of a more coherent work.

Author Response: Following the suggestion, we have revised the introduction to be more focused on the motivation of designing the division of labor for biological upcycling, the key message of the paper. We also elaborated the necessary contexts and specified the connection among different parts of the work. We believe that the revised introduction is more cohesive and sets a better stage for presenting our results.

- In some cases, the conclusions are not completely supported by the experimental data presented, but represent more the starting hypothesis: for instance, in page 7, the authors argument that “by assigning the pathways to separate strains” this leads to the advantage of “carrying a lower metabolic burden”. Even though since is what is expected in theory, not all data can be explained in this way and this should be then critically discussed. It could be argued, for instance, that a lower metabolic burden should then lead to more energy and carbon available for biomass production for the synthetic consortium. However, in Fig 2d) and 2e) the co-culture Pp-T + Pp-E shows a lower biomass production (max OD = 12 against 15) and the biomass growth curve shows almost a diauxic growth pattern, which is not expected in the synthetic consortium. I would therefore recommend that this part of data interpretation regarding the advantage of the division of labor should be elaborated a bit more, to provide more inn-depth discussions, supported by the data in a more comprehensive way.

Author Response: We thank the reviewer for this critical comment and agree with the reviewer that reduced metabolic burden upon DOL is our theoretical reasoning and analysis. To ensure the rigor and accuracy of the presentation, we have removed this specific statement. Meanwhile, we keep other statements relating to DOL, including faster substrate (pure TPA/EG or hydrolysate) consumption and higher product yield, which were directly supported by experimental results. To better understand different biomass accumulations in Figs. 2d and 2e, we repeated the fermentation experiments and quantified metabolite production. Our results (Supplementary Fig. 5c,k) showed that the T-E consortium indeed catabolized TPA and EG faster than the single strain Pp-TE but, meanwhile, it yielded more succinate than Pp-TE. This finding suggested that succinate production likely contributed to reduced biomass accumulation by the T-E consortium than Pp-TE. The results also suggested that redirecting succinate flux to desired outlets such as biomass accumulation or product synthesis is a promising window for future optimization and investigation. In the revised

paper, we reported the new experimental results and added an associated discussion about the findings.

- **Experimental plan:** In most experiments, pure EG and TPA is used and in some cases the real PET hydrolysate is also used to evaluate the effect of inhibiting compounds. It is very important and highly relevant to use directly the diluted crude product from the hydrolysis, in view of a real bioupcycling process that can be scaled up in the future. On the other hand, the real PET hydrolysate was not tested in all experiments and that might lead to some confusion in the reader. It might also be a good idea to underline that when the authors are writing about “TPA” they actually worked with Na₂TPA (at least when using the hydrolysate), which is known to have a better solubility and lower toxicity to the cells compared to pure TPA. Since the authors managed to work with 100 mM TPA I would guess that was also disodium terephthalate? It might be worth clarifying, because in several cases I was in doubt whether pure TPA was used (as stated in the text) or rather Na₂-TPA.

Author Response: In our study, pure EG and Na₂TPA were initially used as substrates to validate the constructed strains (e.g., Figs. 1 and 2). Later, the real PET hydrolysate was utilized to test the feasibility of the consortium for assimilating actual plastic hydrolysate (Fig. 3). Similarly, for producing mcl-PHA and MA (Figs. 4 and 5), we used pure EG and Na₂TPA for strain validation and actual hydrolysate for the demonstration and comparison of product production. In the revised paper, we have added explicit descriptions in all related contexts to avoid confusion

Additionally, as the reviewer mentioned, we indeed used Na₂TPA, instead of TPA, as a model substrate due to its better solubility in water. Based on our knowledge, in previous studies, TPA solution was made similarly by dissolving in NaOH or adjusting pH with NaOH to convert to Na₂TPA and did not specifically name TPA as Na₂TPA (Werner et al., 2021; Kim et al., 2019; Kenny et al., 2012; and Narancic et al., 2021). In fact, this was one of the reasons we used “mM” as units for the titer of TPA to avoid the confusion. To clarify this point, we have emphasized it when TPA is first mentioned in the main text and also emphasized it in the Method section.

References:

- Werner, Allison Z., et al. "Tandem chemical deconstruction and biological upcycling of poly (ethylene terephthalate) to β -keto adipic acid by *Pseudomonas putida* KT2440." *Metabolic Engineering* 67 (2021): 250-261.
- Kim, Hee Taek, et al. "Biological valorization of poly (ethylene terephthalate) monomers for upcycling waste PET." *ACS Sustainable Chemistry & Engineering* 7.24 (2019): 19396-19406.
- Kenny, Shane T., et al. "Development of a bioprocess to convert PET derived terephthalic acid and biodiesel derived glycerol to medium chain length polyhydroxyalkanoate." *Applied microbiology and biotechnology* 95 (2012): 623-633.
- Narancic, Tanja, et al. "Genome analysis of the metabolically versatile *Pseudomonas umsongensis* GO16: the genetic basis for PET monomer upcycling into polyhydroxyalkanoates." *Microbial biotechnology* 14.6 (2021): 2463-2480.

- In the paragraph dedicated to the Upcycling of PET hydrolysate into mcl-PHA, I would recommend mentioning also that there are already studies that have used *P. putida* to upcycle PET monomers into mcl-PHAs. Else it seems that this as this is the first study looking into this, while it has been investigated already for 10 years and there are start-up companies from

universities already scaling up the technology, using engineered *Pseudomonas* strains.

Author Response: We are very grateful for this suggestion. Following the recommendation, we have revised the beginning part of the “Upcycling of PET hydrolysate into mcl-PHA” section to include related contexts and added associated citations as follows:

“Specifically, we selected mcl-PHA, ranging from C6 to C14, as our target product because mcl-PHA is one of the most promising biodegradable polymers and the chassis organism *P. putida* has been shown to produce mcl-PHA from different substrates including sugars, organic acids, aromatic compounds and PET monomers^{45,46}. For instance, engineered *P. putida* KT2440 was reported to produce 372 mg/L of mcl-PHA from pure EG under batch fermentation⁴¹ and its relative, *Pseudomonas umsongensis* GO16, yielded 2349 mg/L of mcl-PHA from pure TPA in fed-batch fermentation⁴⁷ (Supplementary Table 3). However, the titer of *P. umsongensis* GO16 was dropped down to 150-210 mg/L when PET hydrolysate was used as cellular substrate^{17,48}, highlighting the need to investigate the co-assimilation of TPA and EG in the hydrolysate and explore alternative engineering strategies.”

- To highlight better the relevance of the experimental findings in terms of TPA and EG upcycling into the different high-value compounds (and facilitate comparisons with literature), I would recommend presenting a table with the best results expressed in Cmol and the %Cmol that from the substrate ends up into the final product. The fact that an engineered strain is able to perform a new metabolic pathway is of course a very relevant results from a synthetic biology point of view but its real impact from the plastic upcycling point of view depends also on final titers and conversion efficiencies. Since not all the readers are necessarily experts in plastic bioupcycling, this would help underlining the relevance of your results and will also facilitate the discussion part, where the authors can compare their findings against the literature findings.

Author Response: We agree that a table summarizing the best results from our study and previous work would help to elucidate the experimental findings in terms of TPA and EG upcycling and facilitate the comparison with the existing literature. Additionally, considering that most of the references report their results in the unit of g/L or mol/L, we have updated the units of our results to g/L (product mass/culture volume), g/g (product mass/substrate mass) or mol/mol (product molar mass/substrate molar mass). The corresponding results are now provided in Supplementary Tables 3 and 4 of the revised paper, which serve as a basis for comparing mcl-PHA and MA production from this study and previous work.

- In the paragraph regarding muconic acid biosynthesis via DOL, it seems like the experimental design (of the synthetic consortium) is very different compared to the previous sections, so the rationale behind this choice should be clarified better. I would also suggest clarifying more in detail which experimental results demonstrate crosstalk reduction in your synthetic consortium compared to the single engineered strain. I would also suggest checking more thoroughly the latest literature on this topic: MA biosynthesis has also been investigated using TPA and not only with glucose feeding. In general, a more careful use of literature would be appropriate.

Author Response: Prior to presenting the MA biosynthesis, we demonstrated that, through the division of labor (DOL) for TPA and EG consumption, ecosystem design can reduce catabolic crosstalk and facilitate substrate consumption compared to its single strain counterpart, particularly

when substrate concentrations are high or crude hydrolysate was used (Supplementary Fig. 6, Fig. 2, 3). In addition, we also showed that specialized assimilation of TPA and EG can confer a higher level of mcl-PHA production (Fig. 4). In the MA synthesis section, we aimed to demonstrate the unique advantage of DOL in enabling simple and yet effective optimization for the synthesis of desired product from the PET hydrolysate as the sole carbon source. Indeed, we demonstrated that, by partitioning MA synthesis into two strains, we achieved the production optimization by altering initial ratio or inoculation time (Fig. 5). Following the reviewer's comment, we amended the rationale of this MA section.

Additionally, we have searched the latest literature and discussed the related studies as part of the background of the section as detailed below: "Recently, different efforts have been made for MA production (Supplementary Table 4). For example, a study reported a three-step process, involving PET hydrolysis, TPA and EG separation, and microbial catalysis, to convert TPA to MA; however, the resulting MA production was relatively low (0.38 g/L)⁵⁴. Studies also showed a higher final titer of MA synthesis (4.59 – 73.8 g/L) by engineered *P. putida* but, in these cases, glucose was required as a sole substrate or as a supplementary carbon source^{52,55-58}."

- To sum up, I would say that the authors did a tremendous work from a synthetic biology point of view, while the real impact on the plastic upcycling topic should be highlighted better. The authors could underline more the relevance of their findings, showing how the obtained titers and productivities compare to the requirements for a viable process that can be scaled up in the future (and other studies proposing other bioprocesses for the production of IA or PHA)?

Author Response: As suggested, we have expanded the Conclusions section to provide more in-depth discussion about the meaning of our results toward actual plastic upcycling, in addition to summarizing its significance for the field of synthetic biology. Specifically, we have added and elaborated the insights of the study into actual process design and optimization with engineered microbial consortia for enhancing the viability of PET upcycling. The discussion has now been added to the Conclusions section.

- In section regarding the PHA and itaconic acid production, the rationale of this co-production is not very clear to me, as itaconic acid has a market value that can be 10-20 times higher than mcl-PHA. So what would be the point to direct precious carbon to produce PHA instead of maximizing itaconic acid, from a process point of view? This should be clarified so that the reader does not get impression of a mere (high level) synthetic biology "exercise" that, though of high quality, has no real reason or application /impact on the real plastic waste upcycling topic. What is the division of labor advantage in this experiment, in terms of substrate conversion into the 2 products?

Author Response: Following the reviewer's advice of "less can be more", we decided in the revised manuscript to remove the PHA and IA co-production section while spending more efforts on elaborating motivation, related research contexts and discussion on other parts of the results to make the paper more focused and coherent. That being said, the following is our response to this comment: The central focus of our project is to demonstrate the power of microbial division of labor (DOL) in biological PET upcycling. Through the earlier results in the Results, we have showed that synthetic consortia enable more efficient substrate utilization and higher product production than single strains. Here, the objective of the mcl-PHA and IA co-production was to

demonstrate the versatility of the engineered consortia for complex and advanced bioproduction, in addition to reaffirming the previously demonstrated advantage (i.e., the faster substrate consumption, higher production and flexible optimization).

[Redacted]

Finally, we want to reiterate that we removed from the revised manuscript the co-production section while elaborating the motivation, related contexts and discussion of other results to improve the overall clarity and coherence of the paper.

- Furthermore, the downstream processing for PHA is a well-known bottleneck exactly because it is intracellular and requires multiple steps to remove from the cells and purify, without changing the thermoplastic properties. On the other hand, IA is secreted and this would be the real simple downstream separation, from a process point of view. So what is the downstream advantage to coproduce PHA when IA could be continuously produced and easily separated?

Author Response: We appreciate and value the reviewer's feedback. We fully agree with the reviewer that mcl-PHA and IA are produced and stored in separate compartments and thus can be easily separated. In fact, that was the exact reason that we chose to express these two target chemicals, instead of others (e.g., IA and MA). Meanwhile, we want to reiterate that, after careful consideration, we removed the co-production section from the revised manuscript and focused on a more detailed and thorough explanation of the rest results and new data to improve the overall clarity, cohesiveness and readability of the work.

- On a different note: Please use consistent units for the titer of your products. Presenting PHA in mg/L and IA in mM does not facilitate comparison.

Author Response: As requested, we now unified the units for our products presented in the paper. Additionally, we added Supplementary Tables 3 and 4 for a better comparison of mcl-PHA and MA production with previous studies.

- Also: please explain what you mean with "the titer was slower"? (a titer does not contain a time unit).

Author Response: With all due respect, we searched the manuscript but did not find the exact phrase of "titer was slower". Nevertheless, we proofread the revised manuscript to ensure its accuracy.

- In the Conclusion paragraph, a deeper discussion the meaning of these findings for the division

of labor and the relevance for PET upcycling into MA, PHA and IA would be highly relevant, to better underline the contribution of this study to the advancement beyond the state of the art.

Author Response: We thank the reviewer for this constructive comment. As suggested, we have expanded the Conclusions section to provide more in-depth discussion about the insights of the findings in our study. In addition to discussing the conceptual significance of the work, we highlighted the insights into the microbial division of labor for the design and optimization of actual processes for PET hydrolysate consumption and the upcycling into different products that will eventually enhance the viability of PET upcycling.

- I am also a bit worried about the soundness of some conclusions, which are not completely supported by experimental data (and if they are, you might consider explaining it more): for instance, I am also not totally clear if the conclusion about the reduced catabolic crosstalk is based on your experimental results (and if yes, which ones exactly?) or rather the initial hypothesis? Furthermore, is the production of 0,6 mM IA relevant enough to be considered as a coproduction (compared to 200 mM of PHA)? 0.6 mM is in any case rather insignificant for the production of a platform chemical.

Author Response: In response to this comment, we have checked the manuscript and revised our statements as needed to enhance the soundness of the conclusions. The following is our detailed response to the two specific comments (crosstalk reduction through DOL, and mcl-PHA-IA co-production) that the reviewer made:

(1) Reduction of catabolic crosstalk through DOL. Our engineered T-E consortium is composed of two specialists, Pp-T and Pp-E, with each containing only one of the TPA and EG metabolism pathways. By contrast, the single strain Pp-TE contains both TPA and EG assimilation pathways. Our comparative experiments (Fig. 2f and Supplementary Fig. 6 (previously Supplementary Fig. 5)) showed that, in the presence of a high concentration of the substrates (e.g., 100 mM TPA and 100 mM EG), the T-E consortium managed to assimilate TPA and EG almost simultaneously. By contrast, for the single Pp-TE strain, there was a clear diauxic fermentation pattern with EG being depleted first and then TPA consumption, which resulted in a longer time for substrate depletion. This difference provided a clear illustration of the reduction of catabolic crosstalk in engineered consortium than in the generalist. Additionally, in our efforts for mcl-PHA production, we demonstrated that the monoculture-based fermentation failed to deplete EG whereas the consortium fermentation was able to convert both TPA and EG (Fig. 4g). Consistent with our finding, incomplete and suppressed EG assimilation was reported in previous studies when single engineered *P. putida* strains was used to assimilate PET hydrolysate—which contains both TPA and EG and, accordingly, EG was continuously accumulated through the course of fermentation (Werner et al., 2021; Tiso et al., 2021; and Liu et al., 2022).

(2) mcl-PHA-IA co-production. As mentioned in our previous response, we took the reviewer's advice of "less can be more" and removed this co-production section from the revised manuscript while spending more efforts on elaborating motivation, related contexts and discussion on other parts to improve the clarity, focus and coherence of the paper.

[Redacted]

Thus, although IA production in our study was not the highest, the experiment served the purpose of proof-of-concept demonstration.

References:

- (1) Werner, Allison Z., et al. "Tandem chemical deconstruction and biological upcycling of poly (ethylene terephthalate) to β -ketoacidic acid by *Pseudomonas putida* KT2440." *Metabolic Engineering* 67 (2021): 250-261.
- (2) Tiso, Till, et al. "Towards bio-upcycling of polyethylene terephthalate." *Metabolic engineering* 66 (2021): 167-178.
- (3) Liu, Pan, et al. "Valorization of polyethylene terephthalate to muconic acid by engineering *Pseudomonas Putida*." *International Journal of Molecular Sciences* 23.19 (2022): 10997.

- So in conclusion, I agree with the great potential of this approach but this needs to be also supported more clearly by the experimental results, which need to be comparable with other biotechnological solutions. So comparison and critical discussion against the state of the art is essential and not sufficiently addressed in this work.

Author Response: We appreciate the reviewer's reiteration on the need to compare our results with existing efforts and discuss the significance of our work against the state of the art. As we responded previously, comparison with other studies and critical discussion have been added to the revised manuscript, including Supplementary Tables 3 and 4, as well as associated text throughout the paper.

- Regarding the use of literature: Several of the most relevant studies using PET monomers for upcycling to PHA with *Pseudomonas* are missing. Also the comparison with other bioprocesses that upcycle PET into MA is missing, as well as bioprocesses for the production of IA from waste streams. In my view, this is reflected in a not particularly deep discussion about the relevance of the findings of this study.

Author Response: As requested, we have added Supplementary Tables 3 and 4 to the revised paper to summarize the mcl-PHA and MA productions reported in recent literature and also discussed these studies in the corresponding sections of the paper.

- Regarding the figures: they are very interesting and of good quality, but in order to fit all the results in a same figure, the readability becomes difficult (too crowded and too small fonts inside the figures). Also in the supplementary material, the concentrations of the substrates in the legend inside the figures is really hard to read, due to the small font size.

Author Response: We have now increased the font size for both the main and supplementary figures. Additionally, for some figures, we moved legends to the bottom of the figures to increase their readability.

- In Fig 2d) and 2e) Pp-T + Pp-E shows a lower biomass production (max OD = 12 against 15)

and the biomass growth curve shows almost a diauxic growth pattern, which is not expected in the synthetic consortium. It would be interesting if you could elaborate a bit on this.

Author Response: We thank the reviewer for this comment. To address the question, we repeated the fermentation experiment and analyzed metabolites produced during the fermentation. As we noted previously, our results (Supplementary Fig. 5b,f) showed that the T-E consortium catabolized TPA and EG more efficiently than Pp-TE but, in the meanwhile, it led to transient glycolate accumulation and subsequent depletion and also yielded significantly more succinate at the end of fermentation than the generalist Pp-TE. Thus, the results suggested that succinate production likely contributed to the reduction of biomass accumulation in the T-E consortium and redirecting the succinate flux could be a potential target for future investigation and optimization. In our revision, we reported these new experimental results in the revised paper and added a discussion about the findings.

- In Fig 4j: It does not help to present some of the data in mM and others as mg/L. I would recommend to uniform this to facilitate the comparison of conversion processes from substrate to final product.

Author Response: We now have unified the units of the products in Fig. 4j and other figures. We also added Supplementary Table 3 for a better comparison of mcl-PHA production with other studies.

- Fig 4k: the fed batch shows a higher conversion yield? Could you comment on this, also based on literature findings from other studies?

Author Response: We speculate that one reason for the slightly higher yield (mcl-PHA titer per dry biomass weight) in fed batch was reduced biomass increase during the second round of hydrolysate fermentation. Our results showed that the biomass accumulation of Pp-TEP during the initial and second rounds of hydrolysate assimilation was 7.9 and 6.5, respectively; similarly, the biomass accumulation of the consortium during the initial and second stages was 9.8 and 6.7, respectively. The reduction of biomass increase would lead to a higher yield for the same amount of increase of mcl-PHA. Additionally, reduced biomass production suggested that likely more carbon was directed to the synthesis of other metabolites including mcl-PHA. Another reason was the exaggeration of nitrogen limitation. Due to experimental variations in hydrolysis, we noticed that the actual TPA and EG concentrations in the hydrolysate for the second round of supplementation was a bit higher than those at the first round, which could exaggerate nitrogen limitation that is known to promote mcl-PHA production. These factors collectively led to a higher mcl-PHA production per cell biomass in fed-batch fermentation. In the revised paper, we added a short discussion about the minor increase in conversion yield.

- Also: in your upcycling test with the hydrolyzed PET you use equimolar concentration of TPA and EG. Since you used the raw hydrolysate, does this mean that you obtain an equimolar mixture of TPA and EG from your alkali hydrolysis?

Author Response: Yes. As shown in Fig. 3b, the optimized process yielded almost identical molar amount of Na₂TPA and EG from alkali hydrolysis. However, it should be noted that a small

fraction of Na₂TPA in hydrolysate could be recrystallized after pH neutralization for medium preparation. Therefore, EG molar concentration is slightly higher than that of Na₂TPA, which is observable in the 100 mM hydrolysate case (Supplementary Fig. 7).

- Figure S7: since these are the results of the PHA producing strains it might make more sense to also show the PHA, else it is not really possible to see obtain the conversion yields

Author Response: As suggested, we have included the yields of mcl-PHA per dry cell weight and per substrate weight in Supplementary Fig. 8 (previously Supplementary Fig. 7) to provide a more comprehensive view of the conversion yields.

- Fig S9: could you comment on the difference and the meaning of the results between Fig S9b and 9f? Why is PP-ES not showing any inhibition to 2.5 mM CAT and converts it at the same rate of EG? Is this conversion growth associated?

Author Response: Supplementary Fig. 11b,f (previous Supplementary Fig. 9b,f) showed the CAT-to-MA conversion by Pp-EM and Pp-ES, respectively. Pp-EM was constructed by deleting the *catRBC* operon, responsible for converting MA to biomass, and introducing the tac promoter (Ptac) to substitute the native promoter of the major catechol 1,2-dioxygenase (*catA*) responsible for the CAT-to-MA conversion. By contrast, Pp-ES contains the complete *catRBCA* cluster, allowing the cells to alter its native metabolic regulation to produce biomass production from MA. Thus, we did not observe MA accumulation in the Pp-ES fermentation. Consequently, as Pp-ES has two routes for biomass production, one from EG and another from CAT, Pp-ES can have a faster growth to build more biomass which offers more enzymes to consume CAT and EG than Pp-EM. Additionally, we speculate that *catR*, the native regulator in the *catRBCA* operon of Pp-ES (absent in Pp-EM), could potentially regulate cellular physiology and stress response to be more tolerant to CAT. Notably, echoing our finding, a recent report (Kohlstedt et al., 2018) showed that strong synthetic promoters to control *catA* expression yield could lower the MA production compared to the native regulatory system.

Reference:

Kohlstedt, Michael, et al. "From lignin to nylon: cascaded chemical and biochemical conversion using metabolically engineered *Pseudomonas putida*." *Metabolic engineering* 47 (2018): 279-293.

- Fig S11: the results of the ratio 1:5 (described as the optimum in the text) are not shown even though the batch is presented in the fig 11f

Author Response: The result of 1:5 ratio (Pp-TC: Pp-EM) was the best among the different ratios and was present in Fig. 5g. In the revised manuscript, we have also added it to Supplementary Fig. 13 (previously Supplementary Fig. 11) for an easier comparison.

- Fig S12: the captions should allow to read the figures as a stand-alone, but from the description it is not possible to understand that only one of the 2 strains was added with a delay. I suggest to clarify this point in a few words

Author Response: As suggested, we have elaborated the caption of Supplementary Fig. 14 (previous Supplementary Fig. 12).

[Redacted]

Reviewers' Comments:

Reviewer #1:

Remarks to the Author:

I believe the authors have done an excellent job responding to the concerns of the reviewers. I support publication.

Reviewer #2:

Remarks to the Author:

The revised manuscript by Teng Bao and colleagues has carefully addressed all my concerns and questions.

The authors have provided in-depth explanations, clarified any doubts, and also stream-lined the structure of the article.

In my understanding, the authors have critically considered the raised comments, convincingly providing further reasoning, elaborating on the data analysis, as well as providing new supporting experiments.

I consider therefor that the manuscript was significantly improved and can be accepted in the current form.

As a concluding remark, I would like to add that this was one of my most interesting revision experiences, where the whole process turned into a positive and open scientific discussion with the authors. It was stimulating being part of this process.